# Expressivity of Representation Learning on Continuous-Time Dynamic Graphs: An Information-Flow Centric Review

**Sofiane Ennadir**[1,2*], **Gabriela Zarzar Gandler**[1,2*], **Filip Cornell**[1,2*], **Lele Cao**[1†], **Oleg Smirnov**[1],
**Tianze Wang**[1], **Levente Zólyomi**[1], **Björn Brinne**[1], **Sahar Asadi**[1]
[1] *AI Labs, King/Microsoft*
[2] *KTH Royal Institute of Technology*

Reviewed on OpenReview: *https://openreview.net/forum?id=M7Lhr2anjg*

## Abstract

Graphs are ubiquitous in real-world applications, ranging from social networks to biological systems, and have inspired the development of Graph Neural Networks (GNNs) for learning expressive representations. While most research has centered on static graphs, many real-world scenarios involve dynamic, temporally evolving graphs, motivating the need for Continuous-Time Dynamic Graph (CTDG) models. This paper provides a comprehensive review of Graph Representation Learning (GRL) on CTDGs with a focus on Self-Supervised Representation Learning (SSRL). We introduce a novel theoretical framework that analyzes the expressivity of CTDG models through an Information-Flow (IF) lens, quantifying their ability to propagate and encode temporal and structural information. Leveraging this framework, we categorize existing CTDG methods based on their suitability for different graph types and application scenarios. Within the same scope, we examine the design of SSRL methods tailored to CTDGs, such as predictive and contrastive approaches, highlighting their potential to mitigate the reliance on labeled data. Empirical evaluations on synthetic and real-world datasets validate our theoretical insights, demonstrating the strengths and limitations of various methods across long-range, bi-partite and community-based graphs. This work offers both a theoretical foundation and practical guidance for selecting and developing CTDG models, advancing the understanding of GRL in dynamic settings.

## 1 Introduction

Graph-structured data is prevalent across various domains such as chemoinformatics, bioinformatics, and social network analysis. These domains often require sophisticated machine learning approaches, driving the development of Graph Representation Learning (GRL). At its core, GRL aims to embed graph structures into low-dimensional vector spaces, preserving essential structural and semantic features. This embedding facilitates tasks such as node classification (e.g., music genre prediction (Kumar et al., 2019b)), link prediction (e.g., product recommendation (Wu et al., 2019)), and graph classification (e.g., drug discovery (Kearnes et al., 2016)). A cornerstone of this field is the Message-Passing (MP) paradigm (Gilmer et al., 2017), where nodes iteratively exchange and aggregate information from their neighbors to learn representations.

While much of the research has focused on static graphs (Kipf & Welling, 2017; Veličković et al., 2018; Xu et al., 2019), many real-world applications involve evolving graph structures. Such graphs, termed Dynamic Graphs (DGs), undergo changes over time through node/edge addition/deletion or node feature updates — collectively referred to as *events* (Seo et al., 2018; Bai et al., 2021). DGs can be categorized as Discrete-Time DGs (DTDGs) or Continuous-Time DGs (CTDGs), distinguished by their treatment of temporal granularity. While DTDGs aggregate events into fixed time intervals, CTDGs accommodate temporally irregular events,

---

*Equal contribution. Source code: `https://github.com/king/ctdg-info-flow`
†Corresponding author. Email: `lele.cao@king.com` or `lelecao@microsoft.com`.

offering a more flexible and widely adopted representation for DGs. Learning on CTDGs presents unique challenges, such as capturing temporal dependencies and handling sparse, irregular event data.

Many methods have been proposed to address these challenges, often extending the MP framework to incorporate temporal dynamics or novel architectures. However, these methods typically require large amounts of training data to achieve satisfactory performance. This poses a significant limitation in domains like recommender systems, where the graph is often sparse, and observed interactions (e.g., purchasing events) represent a small fraction of possible connections. Under such conditions, models risk overfitting and struggle to generalize effectively.

To address these limitations, Self-Supervised Representation Learning (SSRL) has emerged as a promising paradigm, leveraging unlabeled data to construct auxiliary pretext tasks for model training. In the context of CTDGs, SSRL tasks often derive supervision from the temporal sequence of events, such as predicting the next link or interaction. This approach mitigates the reliance on labeled data and enables models to learn robust representations. Despite its potential, SSRL for CTDGs remains underexplored compared to its adoption in other areas like Computer Vision (CV) and Natural Language Processing (NLP).

This paper presents a review of methods for learning representations of CTDGs with an emphasis on SSRL approaches. While previous surveys have examined DG representation learning (Gravina & Bacciu, 2024; Kazemi et al., 2020), they often focus on general methods without diving deeply into the unique challenges and opportunities associated with applying SSRL to CTDGs. Our work fills this gap by offering a categorization framework based on their theoretical capability to represent specific families of graphs, particularly those commonly encountered in specialized domains. For instance, in recommender systems, the user-item interaction graph is typically a sparse bipartite graph exhibiting a power-law degree distribution (Lü et al., 2012), whereas social networks often form denser small-world networks characterized by high clustering coefficients and short average path lengths (Watts & Strogatz, 1998).

We begin by introducing a general formulation for CTDG and the associated learning methods. Building upon the concept of Information Flow (IF) in MP Neural Networks (MPNNs) (Gutteridge et al., 2023), we derive a theoretical framework that quantify their *expressivity* — the ability to capture and propagate information across temporal events. Focusing on approaches that leverage inherent graph dynamics, we exclude methods relying solely on sequence modeling techniques such as (Du et al., 2016; Sun et al., 2019). The theoretical analysis of expressivity enables us to categorize existing approaches based on their architectural components and the structural properties of the graphs they are best suited for. We then provide a comprehensive review of these methods, structured according to this categorization, with a particular emphasis on SSRL approaches for CTDGs. Finally, we validate our theoretical insights through experiments, offering empirical results on a representative subset of the reviewed methods. Our contributions are as follows:

- **Theoretical expressivity framework**: We propose a framework for analyzing the expressivity of CTDG methods, with a focus on the upper bounds of IF in temporal MP schemes.

- **Method categorization and review**: We review and categorize existing methods based on their performance with specific families of graphs from our theoretical insights.

- **Empirical validation**: We empirically demonstrate the validity of the derived bounds and illustrate key insights through evaluations of several representative methods.

## 2 Theoretical Expressivity Framework

A static graph is defined as $G = (V, E)$, where $V$ is the set of vertices and $E$ the set of edges. Let $n = |V|$ and $m = |E|$ denote the number of vertices and edges, respectively, and $\mathcal{N}(u) = \{v \colon (v, u) \in E\}$ the set of neighbors of a node $u \in V$. The degree of a node $u$ is $|\mathcal{N}(u)|$. A graph is often represented by its adjacency matrix $\mathbf{A} \in \mathbb{R}^{n \times n}$, where the $(i, j)$-th entry denotes the weight of the edge between the $i$-th and $j$-th nodes, or 0 if no edge exists. Node features are represented by $\mathbf{X} \in \mathbb{R}^{n \times D}$, where $D$ is the feature dimensionality, and the $i$-th row of $\mathbf{X}$ corresponds to the features of the $i$-th node.

For DGs, the graph at time $t$ is represented as $G_t = (V, E_t)$, where $E_t$ is the set of edges at time $t$. Events at time $t + 1$ can alter the topology by adding or removing edges or nodes, resulting in $G_{t+1} = (V, E_{t+1})$. To maintain consistency, we treat the node set $V$ as constant over time by assuming nodes with zero degree exist at the start and can transition to active participation as events unfold.

The upcoming formulation will focus on scenarios where the temporal evolution of the graph is primarily driven by the irreversible addition of new edges, as on e-commerce platforms where purchasing events become a permanent part of the system's knowledge. In a reversible scenario like social networks, edge-removal events such as "unfriending" can be modeled as adding an edge with a negative or reverse signal, as this action, combined with the earlier "friending" behavior, conveys information distinct from simply deleting the edge. This perspective ensures that both positive (edge additions) and negative (edge removals) interactions are captured in this work as key mechanisms underlying real-world data-generating processes.

## 2.1 Key Components of CTDG Representation Learning

To address the dynamic nature of CTDGs and their inherent temporal aspects, several recent works, such as TGN (Rossi et al., 2020a), TGAT (Xu et al., 2020), and CTAN (Gravina et al., 2024), have proposed adapted MP mechanisms. They allow nodes to communicate and update their states based on both current events and historical interactions. Specifically, for each node $u$, its temporal neighborhood $\mathcal{N}(u, t)$ at time $t$ is defined to include all nodes that have interacted with $u$ within a specified time window (also referred to as the context window) leading up to $t$. This temporal neighborhood captures the temporal aspect of interactions, ensuring that node updates consider both recent and relevant historical information.

We intend to propose a general framework that encompasses the various approaches relying on temporal MP to facilitate learning on CTDGs. This framework is built on two essential components:

(i) **Node representation**: At any given time $t$, each node $u$ has a representation $h_u(t)$ in the embedding space. This representation is used for various pretraining tasks or downstream applications.

(ii) **Node memory**: Each node $u$ maintains a memory state, denoted as $s_u(t)$, which evolves over time to track the node's historical interactions. While some methods leverage this memory explicitly, others rely solely on the node's current representation.

When an event $\mathcal{E} = (u, v, e_{u,v}^t)$ occurs at time $t$ between nodes $u$ and $v$, where $e_{u,v}^t$ captures the event's features, updates are performed in two stages:

(i) **The nodes directly involved in the event** (i.e., $u$ and $v$) are updated to reflect the new interaction. The generated message directly updates their corresponding memory based on the event's features $e_{u,v}^t$ and also their previous memories in time $t^-$. This can be written as:

$$s_u(t) = \text{MemUpd}([s_u(t^-), s_v(t^-), t - t^-, e_{u,v}^t]); \; s_v(t) = \text{MemUpd}([s_v(t^-), s_u(t^-), t - t^-, e_{u,v}^t]) \quad (1)$$

where function MemUpd varies across methods; some use GRUs or LSTMs (Kumar et al., 2019a), while others opt for a simple identity function (Rossi et al., 2020a).

(ii) **All other nodes** are updated through a MP mechanism designed to propagate the newly introduced information across the graph. This ensures that the entire network incorporates the updated information, extending beyond the immediately affected nodes. We begin by defining the concept of *temporal neighborhood* of each node $u$ at time $t$ as:

$$\mathcal{N}(u, t) = \{(v, e_{u,v}^{t'}, t') \mid \exists (u, v, t') \in G_t\}, \quad (2)$$

where $\mathcal{N}(u, t)$ captures all nodes $v$ that have interacted with $u$ over time, along with the corresponding interaction features $e_{u,v}^{t'}$ and timestamps $t'$. Based on this temporal neighborhood, the representation of node $u$ is updated as follows:

$$\tilde{h}_v^{(\ell)}(t) = \text{Agg}^{(\ell)}(\{\!\!\{(h_u^{(\ell-1)}(t), t - t', e) \mid (u, e, t') \in \mathcal{N}(v, t)\}\!\!\}) \quad (3)$$

$$h_v^{(\ell)}(t) = \text{Update}^{(\ell)}\left(h_v^{(\ell-1)}(t), \tilde{h}_v^{(\ell)}(t)\right), \quad (4)$$

where AGG is a permutation-invariant function, such as summation or attention, that maps node $u$'s neighbors to an aggregated vector, which is passed to the UPDATE function, producing the updated representation for $u$. Different variants of these functions have been studied in the literature, such as graph convolution (GCN) (Kipf & Welling, 2017), or an attention-based mechanism (Veličković et al., 2018), where each considered neighbor is aggregated based on a learned attention value.

In addition to MP components, a *temporal projection* framework is introduced to account for phases of inactivity when a node is not involved in any event. This ensures the representation of inactive nodes continue to evolve over time, capturing the temporal dependencies critical for downstream tasks. The temporal projection can take various forms, such as a simple Multi-Layer Perceptron (MLP) (Kumar et al., 2019a), an attention-based module (Rossi et al., 2020b), or other temporal modeling schemes. For simplicity in our theoretical analysis, an MLP-based projection is often assumed, where the time difference and the current representation are used to update the node's representation.

## 2.2 Expressivity Framework Based on Information Flow (IF)

The aforementioned update scheme illustrates that a node's representation in DGs evolves temporally, with its trajectory heavily influenced by the composition and activity level of its local neighborhood. Nodes that are consistently involved in events or positioned within close proximity to evolving nodes demonstrate particular dynamic representations over time, highlighting the interdependent interplay between network structure and temporal dynamics in shaping node embeddings. In this study, we aim to extend our understanding from a rigorous theoretical perspective. Our primary objective is to elucidate the precise mechanisms by which individual components of the update scheme influence node representations. By decomposing the update process and analyzing its components, we seek to develop a comprehensive mathematical framework that captures the essence of temporal node embedding evolution. This theoretical framework serves multiple purposes. Firstly, it provides insights into how different methods for CTDG representation learning generate distinct node embeddings. By identifying the key factors that drive these differences, we can better understand the strengths and limitations of these methods. Secondly, the framework enables us to predict and explain the performance of different methods across various graph types (e.g., sparse, homogeneous) and applications (e.g., recommendation systems, social networks).

The traditional Weisfeiler-Lehman (WL) framework, which was originally built on ideas from the Weisfeiler-Lehman isomorphism test, and further generalized to higher order variants (Morris et al., 2017), has attracted a lot of attention to modelize graph classifier's capabilities Morris et al. (2019). This approach was recently extended to temporal graphs as temporal-WL (Souza et al., 2022), which iteratively refines node color using node/edge features and timestamps to distinguish non-isomorphic graphs. While effective in some scenarios, the temporal-WL struggles to capture the progression of information over time, particularly the differential impact and propagation of events across snapshots. Its reliance on specific node/edge features further limits its generalization in high-dimensional, real-world settings. This limitation renders the approach not adapted for our aim to understand the effect of each model's components on a node's evolving representation. In contrast, the IF view focuses on how events propagate through consecutive graph snapshots, modeling their influence on node/edge representations over time. By examining embedding changes, this view provides a more nuanced understanding of temporal dependencies and the evolution of graph structure, offering practical insights into the applicability of different techniques in dynamic, event-driven contexts.

Specifically, our goal is to examine the change in node embeddings between consecutive time points to understand how events impact the node representations. Naturally and intuitively, the nodes directly involved in the events are expected to undergo the most significant changes in their embeddings, while the representations of other nodes are less affected. This analysis provides valuable insights into how the graph's structure evolves in response to events, offering a detailed view of how different parts of the graph react. By studying the evolution of node embeddings, we can also assess how various methods handle information propagation across the graph. This sheds light on the strengths and limitations of different techniques, offering guidance on their applicability to specific use cases where certain types of event-driven changes are more prevalent.

Given a graph-based function $f : (\mathcal{A}, \mathcal{X}) \to \mathcal{Y}$, which produces embeddings from dynamic graph snapshots, we assess the quantitative differences $d_{\mathcal{Y}}(f_u(G_{t-1}), f_u(G_t))$ using a distance metric $d_{\mathcal{Y}}$ in the embedding space. Based on the graph node distribution $\mathcal{D}_V$, we define the *flow quantity* as:

$$\mathcal{F}_u[f] = \mathbb{E}_{u \sim \mathcal{D}_V}[d_{\mathcal{Y}}(f_u(G_{t+1}), f_u(G_t))].$$

This *flow quantity* is influenced by both the graph function $f$ and the node distribution of the $t$-th DG snapshot. While the exact computation of $\mathcal{F}_u[f]$ is challenging due to the dynamic nature of the graph, we can approach it from an upper-bound perspective efficiently. Notably, this analysis assumes the degree distribution of nodes remains relatively stable between consecutive time snapshots, as changes in degree distribution primarily occur over longer intervals. This stability assumption simplifies the derivation, allowing us to focus on localized structural changes in CTDGs.

**Definition 2.1** (Continuous IF). *Let's consider the CTDG-based function $f : (\mathcal{A}, \mathcal{X}) \to \mathcal{Y}$, we say that node $u$ is $\gamma$-flowing if $\mathcal{F}_u[f] \le \gamma$.*

We consider $\|\cdot\|$ as a suitable $p$-norm that induces a distance metric $d_{\mathcal{Y}}(\cdot, \cdot)$ in the output embedding space $\mathcal{Y}$. Using this definition, we aim to compute an upper bound, denoted as $\gamma$, which quantifies the potential variation in a node's representation following an event. This upper bound serves as an indicator of how much a node's embedding may be affected by modifications in the graph's structure, providing a measure of the sensitivity of the representation to such events. Since all $p$-norms in finite-dimensional spaces are equivalent up to a multiplicative constant, the choice of a specific norm $\|\cdot\|_p$ affects the upper bound $\gamma$ only by such a constant. Throughout our analysis, $\|\cdot\|$ denotes the Euclidean (resp., spectral) norm.

**Theorem 2.2** (GCN-based aggregation). *Let's consider a CTDG-based function $f : (\mathcal{A}, \mathcal{X}) \to \mathcal{Y}$ based on $L$ GCN-like layers. After an event between node $i$ and another node, the following properties hold for any node $u$ not involved in the event:*

- When $L < d(u, i)$, we have $u$ is $\gamma$-flowing with

$$d_{\mathcal{Y}}(f_u(G_{t+1}), f_u(G_t)) \le \hat{w}_u \|W_t\| \prod\nolimits_{l=1}^{L} \|W^{(l)}\|;$$

- When $L \ge d(u, i)$, we have $u$ is $\gamma$-flowing with

$$d_{\mathcal{Y}}(f_u(G_{t+1}), f_u(G_t)) \le \prod\nolimits_{l=1}^{L} \|W^{(l)}\| \big[ \hat{w}_u \|W_t\| + \hat{w}_{u,i} \Delta_{t,t+1}(s_i) \big],$$

  where $\hat{w}_u$ is the sum of temporal normalized walks of length $(L-1)$ starting from $u$; the term $\hat{w}_{u,i}$ denotes the normalized shortest path between $u$ and $i$; the terms $W_t$ and $W^{(l)}$ are weight matrices involved in the propagation, with $W_t$ capturing temporal updates; $\Delta_{t,t+1}(s_i)$ is the difference in memory state for node $i$, as introduced by the event.

In a nutshell, Theorem 2.2 provides an upper bound with respect to Definition 2.1 for two distinct cases. The first occurs when the node $u$ is outside the $L$-hop neighborhood of the node involved in the event, and the second occurs when it is within the neighborhood. This distinction has practical implications: in some instances, such as gene mutation, the next event is likely to occur within the same neighborhood, while in others, like recommender systems, it may happen in a completely different part of the graph. Consequently, understanding how each component affects the evolving node representation is essential for evaluating its applicability to specific scenarios. While Theorem 2.2 focuses on the GCN-based aggregation scheme, these results can also be adapted to more general attention-based aggregation model described in Section 2.1.

**Theorem 2.3** (Attention-based aggregation). *Let's consider a CTDG function $f : (\mathcal{A}, \mathcal{X}) \to \mathcal{Y}$ based on $L$ attention-based layers. After an event between node $i$ and another node, the following properties hold for any node $u$ not involved in the event:*

- When $L < d(u, i)$, we have $u$ is $\gamma$-flowing with

$$d_{\mathcal{Y}}(f_u(G_{t+1}), f_u(G_t)) \le deg(u) \big[ \|W_t\| + B \|W_t\|^2 \big];$$

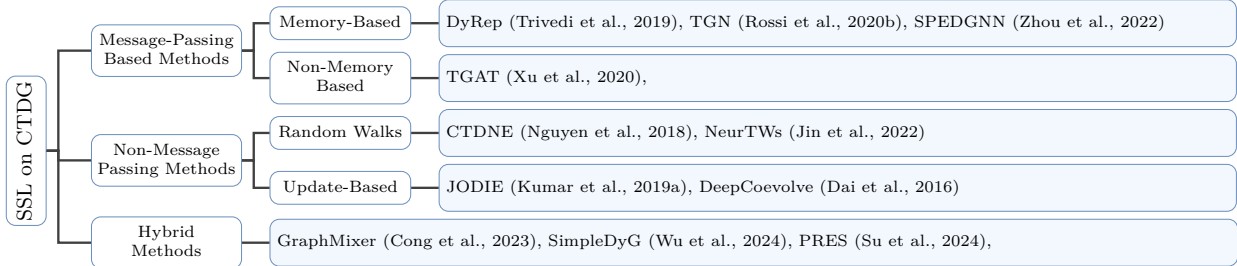

Figure 1: Categorization and taxonomy of GRL methods on CTDGs.

- When $L \geq d(u, i)$, we have $u$ is $\gamma$-flowing with

$$d_{\mathcal{Y}}(f_u(G_{t+1}), f_u(G_t)) \leq deg(u)\big[\|W_t\| + B\|W_t\|^2\big] + \Delta_{t,t+1}(s_i),$$

where $deg(u)$ denotes the degree of node $u$; and $B$ is an upper-bound of hidden representation space.

Based on Theorem 2.3, the propagation of information to indirectly involved nodes appears to be influenced by three key factors. The first is the node's connectivity, typically reflected by its degree. This result is intuitive, as highly connected nodes are more likely to fall within the necessary temporal neighborhood and thus receive frequently occurring updates. The second factor is the inclusion of a memory component, which can enhance the flow of information by retaining past interactions that are relevant for future updates. The third factor is the temporal projection mechanism, which updates node representations over time to reflect changes in the graph's structure.

The expressivity upper bounds (cf. Appendix B and C for derivation details) allow us to categorize existing methods in the literature based on whether they incorporate one or more of these components. This categorization provides insights into how node representations evolve and offers guidance on the suitability of different methods for various scenarios, depending on the used aggregation framework (in terms of connectivity), memory, and temporal dynamics.

## 3 Method Categorization and Review

This section reviews GRL methods for CTDGs, categorizing them based on our expressivity framework and exploring how SSRL approaches align with this framework to leverage unlabeled data effectively.

### 3.1 Method Categorization Inspired by Theoretical Framework

The proposed expressivity framework in Section 2.2 provides a systematic basis for evaluating the strengths and limitations of each method across different graph types. Typically, we can see that a model's expressivity depends on the message-passing framework and the usage of memory where relying only on one single-hop update with minimal memory might have a small upper-bound on its IF for distance node, limiting therefore its capacity to melodize the event occurrence. Based on these insights, we introduce a categorization (Figure 1) to distinguish between the different choices. An additional visual summary is provided in Appendix F.

#### 3.1.1 Non-message-passing (Non-MP) methods

One of the earliest approaches to address CTDG is JODIE (Kumar et al., 2019a). The proposed approach captures interaction dynamics between users and items by learning two types of embeddings: static embeddings for long-term characteristics and dynamic embeddings that evolve over time based on interactions. JODIE employs two RNNs (for users and items respectively), whose outputs are interdependent and update upon each arrival of event. To manage inactivity periods, the model incorporates a projection step that adjusts embeddings using an attention-like mechanism. While the items and users can be seen as nodes

within a graph, the method doesn't rely on MP mechanism and only leverages temporal projection to update representations, allowing node embeddings to evolve over time, even in the absence of interactions. Based on Theorem 2.2, we consider that this temporal evolution is useful for predicting recurring events, such as seasonal product demand in recommendation systems. However, we argue that JODIE struggles with stochastic events, especially in multi-community graphs, where sudden, unpredictable changes are more prevalent. Similarly, DeepCoevolve (Dai et al., 2016) also employs two mutually-recursive RNNs to generate embedding trajectories. The key difference is that, unlike JODIE, DeepCoevolve does not use a temporal projection and consequently doesn't update node representations based on temporal dynamics; instead, it maintains a node's embedding consistently between events or interactions.

Another category of None-MP methods rely on Random Walks (RW) to generate node embeddings. Nguyen et al. (2018) proposed a pioneering approach in this area, introducing the concept of temporal RW. In static graphs, RW can be freely generated; but in CTDG, these walks must adhere to the temporal dimension, ensuring that steps occur at non-decreasing time. This approach proposes to select nodes and generate walks within a node's temporal neighborhood, incorporating probability and threshold-cut strategies to ensure the generated walks are representative. These walks are then processed using a variant of the Node2Vec algorithm (Grover & Leskovec, 2016) to produce node representations for various downstream tasks. Further exploration in this field has refined the sampling strategy, such as considering the graph's topology (Jin et al., 2022). While these methods are promising, they fall outside the specific scope of our theoretical analysis, which focuses on methods leveraging MP and temporal dynamics.

### 3.1.2 Message-passing (MP) based methods

Alongside the Non-MP category, methods propagating information across the graph structure were proposed. One such approach is DyRep (Trivedi et al., 2019), which models both the topological evolution of the graph and the temporal dynamics of node interactions. DyRep updates a node's representation by considering its direct neighbors, its previous representation, and a temporal projection technique based on a temporal point process, which treats each incoming interaction event as an observation of a stochastic process. While this method checks many important boxes from our theoretical analysis, there is a notable limitation in its reliance on direct neighborhoods. Nodes beyond one-hop distance (i.e., two or more hops from the event) are updated solely through the temporal projection and non-updated neighbors, without direct MP from further nodes. This can lead to over-smoothing when events are concentrated within the same neighborhood, as the aggregation pulls in redundant information and dilutes useful features over time. As a result, while DyRep performs well in scenarios where one-hop interactions are crucial (such as recommender systems and bi-partite graphs), it is less effective in extracting information from longer-range interactions in larger, sparser graphs. These insights and limitations are also applicable to the similar approach DyGNN Ma et al. (2020), which consists of two main components: an update mechanism that adjusts the representation of nodes involved in events using two LSTMs and a propagation unit that disseminates information within the 1-hop neighborhood of the event nodes.

Xu et al. (2020) extended DyRep and DyGNN by proposing TGAT. They incorporate Functional Time Encoding to map temporal data into a high-dimensional space. They also introduce Temporal Graph Attention Layer to aggregate information from a node's temporal neighbors using a self-attention mechanism that accounts for both structural and temporal aspects of the neighborhood. TGAT does not necessarily employ a memory component, as described in Section 2.1, but instead uses the node features directly as the initial representation $h_u^0$. Therefore, as highlighted by Theorem 2.3, selecting an appropriate number of layers $L$, and thus defining the $L$-hop neighborhood, is crucial for ensuring effective information propagation.

In a similar perspective to TGAT, TGN (Rossi et al., 2020b) offers a general formalization of CTDG functions, akin to the framework we presented in Section 2.1. TGN incorporates both temporal projection and information propagation through attention-based aggregation. Theorem 2.3 is directly applicable to this method, emphasizing the need to adjust the number of layers based on the specific use case. For example, in recommender systems, setting $L = 2$ may suffice, while in social networks or molecular graphs, the appropriate value of $L$ should be determined by the graph's structural properties, such as its diameter or betweenness centrality, which is captured by the degree term in our provided upper-bound. This approach was further explored by Souza et al. (2022), who examined the expressive power of the TGN architecture.

Their work highlighted the importance of the memory component and the advantages of using injective MP functions. Building on these insights, they introduced PINT, which enhances the expressivity of the TGN model by incorporating injective aggregation and update functions, along with augmenting memory states with relative positional features.

The temporal dimension has been further explored in recent work, such as FreeDyG (Tian et al., 2024), which approaches the temporal perspective through the frequency domain. FreeDyG's update mechanism utilizes two LSTMs (for source and target node respectively), along with a merge unit that incorporates information from the direct neighborhood and propagates it to adjacent nodes. Its key innovation lies in the use of the Fast Fourier Transform (FFT) to convert time-domain data into the frequency domain, allowing the model to identify key interaction patterns. The process is then reversed using the inverse FFT to project the data back into the time domain. This approach enables the model to capture temporal patterns in node interactions more effectively. However, the method remains limited by its reliance on the direct neighbors of the nodes involved in an event. Consequently, while FreeDyG can better track various temporal trends in some frequency-structured cases, it still falls short in fully considering the structural aspects of the graph, as discussed earlier. In the same line, SPEDGNN (Zhou et al., 2022) introduces adaptive spectral Transforms and global Framelet graph convolutions to enable efficient long-term dependency modeling and multi-scale graph feature extraction. Unlike TGAT, it avoids discrete time snapshots by encoding temporal and structural information directly in the spectral domain, ensuring scalable and well-conditioned representations.

A common limitation among many existing methods is their sensitivity to the number of layers, which, if not well-matched to the graph's topology, can hinder performance in capturing long-range dependencies. Our theoretical analysis highlights this, showing how the upper bound depends on the topology, either through normalized walks or node degrees. Increasing the number of layers, $L$, does not necessarily enhance long-range dependency modeling and can lead to issues like over-smoothing or over-squashing. This limitation inspired the development of CTAN (Gravina et al., 2024), which addresses information propagation following an event by learning a diffusion function, modeled as a dynamical system governed by a learnable ordinary differential equation. This approach is particularly beneficial for CTDGs, where learning temporal patterns from irregularly sampled timestamps is essential. Similarly, Petrović et al. (2024) proposed Temporal Graph Rewiring, a method designed to facilitate message passing between distant nodes in temporal graphs. By employing expander graph propagation, a technique with proven success in static graphs, this approach enables more efficient long-range interactions in dynamic settings.

### 3.1.3 Hybrid methods

Another category of methods in the literature falls outside of our proposed categorization, adopting a hybrid approach that aims to propagate information within DG without strictly adhering to classical MP schemes.

One primary focus of hybrid methods is addressing the complexity challenges associated with existing CTDG methods. Complexity has two main aspects: computational (time or operation) complexity and storage (space or memory) complexity. From the computational perspective, Cong et al. (2023) analyzed the impact of incorporating RNNs and self-attention mechanisms into MP on model accuracy and performance. While these components can enhance performance, the study revealed that they are not strictly necessary to achieve strong results. So, the authors introduced GraphMixer, a simpler yet highly effective architecture composed of a "link-encoder" based on MLPs, a "node-encoder" employing neighborhood mean pooling, and an MLP-based "link-classifier". This design significantly reduces both computational complexity, as MLPs are considerably more time-efficient than attention-based models and RNNs. Similarly, SimpleDyG (Wu et al., 2024) leverages the Transformer's self-attention mechanism by modeling the dynamic graph as a sequence that captures temporal evolution patterns through an alignment technique. By transforming the graph into a sequence and applying an unmodified Transformer architecture, SimpleDyG demonstrate lower operational complexity, resulting in a sparser and more efficient design.

Another focus of hybrid methods addresses batching strategies, emphasizing the need to preserve time dependencies, particularly when handling larger batch sizes. One notable approach is PRES (Su et al., 2024), which introduces an iterative prediction-correction scheme to mitigate temporal discontinuities, complemented by a memory-smoothing objective to maintain memory coherence. This method achieves comparable performance to previous works while significantly improving speed, enabling the use of larger batch sizes.

### 3.2 Relevance of Self-Supervised Representation Learning (SSRL) Methods for CTDG

While the methods reviewed in the above categories are capable of extracting meaningful node and graph representations for a variety of tasks, they typically require large volumes of labeled data for effective training. However, the exponential growth of CTDG data across diverse fields in domains such as social networks, recommender systems and biological interaction networks, has rendered manual labeling both labor-intensive and prohibitively costly. As a result, SSRL has become an increasingly popular approach to address the scarcity of labeled data. SSRL tackles this challenge by creating alternative, auxiliary tasks from unlabeled data to help models learn useful representations. These auxiliary tasks, often referred to as *pretraining* or *pretext tasks*, allow models to leverage the inherent data structure as a learning signal. By defining these pretext tasks carefully, SSRL enables the model to capture essential features and patterns in the data without needing explicit labels. SSRL has achieved considerable success in fields like CV (YM. et al., 2020; He et al., 2022) and NLP (Radford et al., 2018), forming the backbone of foundational models that underpin today's AI systems (Bommasani et al., 2021).

We would like to highlight that the theoretical analysis provided in Section 2.2, grounded in IF, does not consider the presence of labels. Instead, it focuses on examining how an event influences a node's representation, thereby evaluating the method's expressive power in modeling the CTDG at time $t$. While this analysis has been instrumental in categorizing various methods as in Section 3, it is also highly relevant in the context of SSRL. A method's expressive power is essential for designing effective pretext tasks; for instance, if a method has limitations in propagating information over long ranges, contrastive augmentations that depend on capturing long-range dependencies may fail to deliver robust pretraining outcomes.

In the context of CTDGs, SSRL has significant potential for tasks such as link prediction, anomaly detection, and temporal pattern recognition, where labels are often unavailable or costly to obtain. Additionally, another learning paradigm, semi-supervised learning, has also been explored within the DG context. This approach is particularly useful for node classification tasks where only a subset of nodes is labeled, and the objective is to propagate these labels to the remaining unlabeled nodes. Semi-supervised learning has been widely applied in static graph settings, leveraging known labels to guide the representation learning process for all nodes within the graph. In this work, we focus on SSRL approaches in the context of CTDG. The next section will therefore focus on reviewing various SSRL-based methods within the CTDG context, exploring different approaches for designing pretext tasks that harness the temporal and structural aspects of evolving graphs to learn robust representations.

#### 3.2.1 Pretext tasks for pretraining on CTDG

The commonly adopted pretext tasks for pretraining SSRL model on CTDG can be largely divided into two main categories: **predictive** and **contrastive** pretraining tasks.

**Predictive pretraining** often relies on auto-regressive modeling paradigms, leveraging the sequential nature of temporal data. This task is particularly valuable in domains where data naturally follows a temporal or ordered structure, such as NLP or time-series analysis. In NLP, for example, predicting the next word based on the context of preceding words has been shown to help models learn robust and generalizable representations of language (Kenton & Toutanova, 2019). These learned representations capture semantic and syntactic patterns that enhance performance on a range of downstream tasks. For CTDGs, where the temporal aspect of interactions between nodes is a key feature, predictive pretext tasks are equally relevant. By taking advantage of the temporal ordering of events, an intuitive auxiliary pretext task emerges: predicting the occurrence of the next event. This task involves, at any given time step $t$, forecasting the subsequent event at the following time step $t+1$, underpinning the majority of SSRL approaches for CTDG.

In the context of link prediction within CTDGs, this task often translates to predicting which two nodes will interact in the next event, such as anticipating a user's next purchase or interaction in recommender systems (Kumar et al., 2019b). Such predictions provide critical insights into temporal dynamics and user preferences, enhancing the model's ability to generalize to unseen events. Specifically, it uses observed events to optimize the negative log-likelihood of event prediction (Xu et al., 2020; Rossi et al., 2020b; Trivedi et al., 2019). Observed events are treated as positive instances, while non-occurring events are designated as negative instances, which facilitates a form of binary classification. However, since evaluating all possible

non-occurring events is computationally infeasible in large-scale graphs, negative sampling is commonly employed. Negative sampling draws a manageable set of negative samples from the potential edges based on a predefined sampling distribution, thereby reducing complexity while still providing informative gradients for learning.

**Contrastive pretraining** seeks to learn an embedding space by positioning semantically similar pairs, such as augmented versions of the same input or closely related elements closer together, while pushing apart pairs that are deemed dissimilar, referred to as negative pairs. This process helps the model to learn discriminative features that are useful for differentiating between various inputs in downstream tasks. By optimizing this contrastive objective, the model gains an enhanced ability to capture meaningful feature distinctions and similarities within the data. This task has achieved notable success in domains like CV, where it has significantly advanced performance by leveraging image augmentations to create so-called positive pairs.

However, contrastive pretraining has been less explored in CTDGs. Only a few studies (Chen et al., 2023; Sun et al., 2022) have extended this technique to DG. In particular, DySubC (Chen et al., 2023) introduced a contrastive learning framework designed to maximize the mutual information between each node's embedding and the representation of its associated temporal subgraph. DySubC consists of generating time-aware subgraphs for each node by capturing both structural information from the node's neighborhood and temporal information from edge timestamps, thereby encoding the dynamics of interactions over time. It then constructs positive and negative samples: positive pairs consist of the central node and its time-weighted subgraph representation, while negative samples are created by applying structural and temporal perturbations, such as shuffling the subgraph structure or removing temporal weight edges. In the same line, CLDG (Xu et al., 2023) approaches contrastive learning by leveraging temporal translation invariance through a sampling layer that extracts persistent signals across snapshots. The aim is to maintain a node's local and global representations while mitigating semantic shifts introduced by temporal augmentations. Additionally, IDOL makes use of a specialized Personalized PageRank mechanism to capture topological changes, and integrates a topology-monitorable sampling strategy yielding better representations.

### 3.2.2 Downstream tasks

Pretraining the model with SSRL pretext tasks aims to learn a robust graph function that provides useful representations for a range of predictive tasks, commonly referred to as *downstream tasks* in the literature. In the context of CTDGs, the literature has focused on three primary downstream tasks:

**Link prediction** task is the most studied one that involves predicting future links for a CTDG. That is to predict the next interaction, identifying both the source and target nodes. Link prediction is particularly relevant in recommender systems, where the goal is to anticipate the next item a user may be interested in. By accurately predicting these links, we can enhance recommendation quality and increase the likelihood of user engagement or sales. In the same perspective related to edge, a less explored task in the literature is the one related to edge classification, regression and clustering, the main goal of which is to assign each edge a class/value/cluster at a future time.

**Node-level prediction** mainly refers to node classification/regression in CTDGs, which essentially extends the static node-level prediction tasks by incorporating both the temporal dynamics and topologies. For example, in social networks, node classification may be used to detect spammers or malicious actors by analyzing patterns in their evolving connections and activities. As networks evolve, so do the relationships between nodes, meaning that a node's classification needs to be updated dynamically based on new interactions.

**Graph-level prediction** tasks aim to classify or regress entire graphs rather than individual nodes, making them especially relevant in domains like bioinformatics. For instance, in molecular graph analysis, each graph represents a molecular structure, and the task involves predicting graph properties such as toxicity or stability based on the configuration of the structure. When graphs evolve over time, the objective shifts to assessing whether the graph remains valid under specific criteria. For example, the addition of certain bonds in a molecular graph may result in a chemically viable or unstable structure.

Our theoretical analysis introduced in Section 2.2 primarily focuses on the impact of events on a node's representation, making it particularly well-suited for link prediction tasks. However, it remains applicable to

other tasks as well, as these tasks are also fundamentally based on node representations, typically aggregated using pooling or `ReadOut` functions.

## 4 Empirical Validation

This section presents an empirical study to validate the theoretical insights and categorization proposed earlier. Specifically, we evaluate the effectiveness of reviewed methods and identify the types of graphs where they perform best. We first outline the experimental setup, followed by a discussion of the empirical findings. Additionally, an analysis of the tightness of the proposed upper bounds is provided in Appendix D.

### 4.1 Experimental Setup

Our experimental focus is on the popular link prediction task, chosen as a representative challenge within the broader scope of CTDGs. We examine a diverse selection of models from those reviewed in Section 3, focusing on models that capture different facets of CTDG methodologies: (I) JODIE (Kumar et al., 2019a), which models evolving embeddings over time; (II) DyRep (Trivedi et al., 2019), designed for capturing dynamic node interactions; (III) TGN (Rossi et al., 2020b), i.e., Temporal Graph Networks for inductive learning; (IV) TGAT (Xu et al., 2020), known for its attention-based approach for handling temporal information; (V) CTAN (Gravina et al., 2024), focused on modeling long-range dependencies within graphs. Experimental details and hyper-parameters used to train/test the model are provided in Appendix G.2.

In alignment with the selected methods, we choose a set of both synthetic and real-world DG datasets which are representative to the main application areas and unique characteristics of CTDGs. Additional details on each dataset, including design considerations and properties, are provided in Appendix G.1.

**Long-range graphs** are those requiring long-range reasoning to achieve strong performance, as distant nodes or graph regions can influence each other. To capture these interactions, specific information propagation mechanisms are necessary, as conventional MP often fails to bridge these long-range dependencies effectively. Evaluating performance on these graphs is crucial to assess a method's ability to transfer information across distant parts of the graph. For this setting, we use an approach similar to the one proposed by Gravina et al. (2024) where we consider a temporal version of the PascalVOC-SP graph (Dwivedi et al., 2022). Additional information about the generation is provided in Appendix G.1.

**Bi-partite graphs** play a crucial role in applications like recommender systems, a prominent use case for CTDGs. To evaluate how different methods handle interactions between disjoint node sets, which are critical for recommendation quality, we use the TGBL-Wiki dataset from the TGB benchmarks (Huang et al., 2024). This dataset represents a bi-partite network where nodes are editors and Wiki pages, with edges added when an editor modifies a page at a specific timestamp. Edges carry text features derived from the page edits. Dataset statistics can be found in Table 6 in Appendix G.1.

**Community-based graphs** consist of nodes grouped into distinct clusters, with relatively few inter-community links compared to intra-community connections. A Barbell graph is a representative example of this structure, where inter-community links act as bottlenecks (Alon & Yahav, 2021), making these graphs challenging for temporal MP schemes due to limited information propagation between communities. To evaluate how well a benchmark captures intra-community links, we generate a synthetic dataset using the Stochastic Block Model (SBM) with the NetworkX (Hagberg et al., 2008) package. This involves creating $B$ dense communities with connections randomly sampled based on a normalized density over a time horizon. As time progresses, the graph evolves, becoming increasingly connected across clusters. For evaluation, we leverage the connection sampling densities between clusters to rank nodes across active communities, measuring performance using Normalized Discounted Cumulative Gain (NDCG) (Järvelin & Kekäläinen, 2002). More details on the parameters, generation and evaluation process are provided in Table 5 in Appendix G.1.

### 4.2 Results Analysis

Table 1 presents the average AUC scores along with their corresponding standard deviations. As anticipated and discussed in Section 3, since the graph structure and topology are important for the downstream classification task, MP based models significantly outperform non-MP approaches like JODIE. This specific

use-case showcases the limitation of this latter family of models, which rather focus on updating the evolved nodes in each event. We additionally see that CTAN is outperforming all the other MP based methods, confirming therefore its ability to track long-range dependencies within the graph.

Table 2 summarizes the average test and validation MRR (Mean Reciprocal Rank) scores, along with standard deviations, for various methods on the real-world bi-partite dataset TGBL-Wiki. The results show that JODIE performs similarly to the MP based methods DyRep and TGAT. While JODIE might theoretically be expected to excel on bi-partite graphs, the dataset's low "surprise factor" — a measure of how predictable the interactions between nodes are — limits its advantages. In this setting, capturing long-range temporal and structural dependencies becomes more critical for the downstream task. Consequently, methods like CTAN and TGN, which are better equipped to model such dependencies, achieve superior performance.

Table 1: Average test AUC ($\pm$ denotes standard deviation) of the different methods on long-range PascalVOC 10 and 30 datasets. Best performance per dataset in **bold**.

| Model | Long-Range Graph | |
|---|---|---|
| | **PascalVOC 10** | **PascalVOC 30** |
| JODIE | $0.67 \pm 0.03$ | $0.65 \pm 0.07$ |
| DyRep | $0.69 \pm 0.02$ | $0.70 \pm 0.02$ |
| TGN | $0.78 \pm 0.10$ | $0.71 \pm 0.04$ |
| TGAT | $0.77 \pm 0.02$ | $0.76 \pm 0.03$ |
| CTAN | $\mathbf{0.80 \pm 0.01}$ | $\mathbf{0.78 \pm 0.01}$ |

Similarly, Table 3 presents the average test and validation NDCG scores, along with their standard deviations, for the evaluated benchmarks on the SBM community-based synthetic datasets. The results indicate that CTAN achieves superior performance compared to the other methods, as expected, given its ability to effectively model long-range dependencies, which is critical to facilitate information to flow within the cluster and to capture community structures. Additionally, TGN demonstrates strong performance, surpassing most methods. These findings align with expectations, as the community-based nature of the dataset benefits from the self-attention mechanisms utilized by TGN.

## 5 Conclusion

In this paper, we presented a comprehensive review of Graph Representation Learning (GRL) on Continuous-Time Dynamic Graphs (CTDGs), emphasizing the critical role of expressivity in capturing temporal and structural dynamics. We introduced an Information-Flow (IF) centric theoretical framework to quantify the expressivity of CTDG models, providing a structured analysis of their performance across graph types and application scenarios. Leveraging this framework, we categorized existing methods, highlighting their strengths and limitations in addressing key challenges such as long-range dependency modeling, temporal irregularity, and sparse data. Additionally, we examined the growing relevance of Self-Supervised Representation Learning (SSRL) for CTDGs, offering insights into predictive and contrastive pretraining tasks and their applicability to real-world datasets. Empirical validation supported our theoretical insights, demonstrating the nuanced trade-offs between different methods across synthetic and real-world datasets. By bridging theoretical foundations with practical evaluations, this work provides a robust roadmap for advancing GRL in dynamic settings, enabling more expressive and adaptable models for a broad spectrum of applications.

Table 2: Average Test and Validation MRR ($\pm$ denotes standard deviation) of the different approaches on the real-world Bi-Partite dataset TGBL-Wiki. Best performance in **bold**.

| Model | Bi-Partite Graph | |
|---|---|---|
| | **Test** | **Val** |
| JODIE | $0.261 \pm 0.123$ | $0.325 \pm 0.149$ |
| DyRep | $0.263 \pm 0.005$ | $0.317 \pm 0.008$ |
| TGN | $0.577 \pm 0.015$ | $0.617 \pm 0.009$ |
| TGAT | $0.282 \pm 0.022$ | $0.323 \pm 0.008$ |
| CTAN | $\mathbf{0.611 \pm 0.011}$ | $\mathbf{0.647 \pm 0.012}$ |

Table 3: Average Test and Validation NDCG scores ($\pm$ denotes standard deviation), on the considered benchmarks on the SBM Community-Based synthetic dataset. Best performance in **bold**.

| Model | Community-Based Graph | |
|---|---|---|
| | **Test** | **Val** |
| JODIE | $0.8491 \pm 0.003$ | $0.863 \pm 0.002$ |
| DyRep | $0.811 \pm 0.005$ | $0.819 \pm 0.006$ |
| TGN | $0.8559 \pm 0.0264$ | $0.8572 \pm 0.0273$ |
| TGAT | $0.8459 \pm 0.0001$ | $0.8564 \pm 0.0002$ |
| CTAN | $\mathbf{0.868 \pm 0.002}$ | $\mathbf{0.872 \pm 0.003}$ |

## Acknowledgments

We would like to thank the entire ACE (AI Center of Excellence) at King (a part of Microsoft) for their continuous support, insightful discussions, and collaborative spirit throughout this project.

We express our sincere gratitude to King's OSC (Open Source Council) for their invaluable support throughout the open source release process. Special thanks go to Colin Cashin, Nate Cleveland, Hugo Cura, Paul Jordaan, David Lorenzo, and Raul Pareja for their dedicated efforts in facilitating the publication and code release process. We are also deeply appreciative of King's CTO, Eric Bowman, for his careful review and approval of our code (`https://github.com/king/ctdg-info-flow`).

Additionally, we are grateful for the diligent review of our manuscript from legal and communications perspectives. Special thanks to Alexandra Stark and Tim Elgar for their essential contributions in ensuring alignment with legal and communications standards.

This work was partially funded by the Wallenberg AI, Autonomous Systems and Software Program (WASP). We also acknowledge the collaboration between King/Microsoft and KTH Royal Institute of Technology, which made this research possible.

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

# The Appendix of

## Expressivity of Representation Learning on Continuous-Time Dynamic Graphs: An Information-Flow Centric Review

## A  Preliminaries

**Notations** For our theoretical analysis, we consider the following notations:

- $d(u, v)$: is the shortest path distance between node $u$ and node $v$.

- deg(u): denotes node $u$'s degree, i.e $|\mathcal{N}(u)|$.

- $d_{\mathcal{Y}}$ is the distance within our output manifold $\mathcal{Y}$.

- The node involved in the event is denoted as $i$.

- $L$ denotes the number of layers used in the considered model.

**Problem Set-up.**  As our goal is to understand the effect of each component on the model's evolving dynamics, we consider the most general use-case of a CTDG function. Typically, we consider that our model $f$ follows the proposed components in Section 2.1. Specifically, we consider the model to based on the described temporal message-passing (either through a GCN-like aggregation or attention-based). We also consider that it makes use of the memory state and not only the node's individual representation. We additionally consider that the model encompasses a Linear temporal projection function to take into account temporal inactivity.

**Theoretical Assumptions.**  For our theoretical analysis, we assume that all models use 1-Lipschitz continuous activation functions, which covers the majority of commonly used functions, including ReLU, LeakyReLU, and TanH (Virmaux & Scaman, 2018). We also focus on the effect of a single event, in contrast to the more practical setting where a batch of events is considered. Under this single-event assumption, the node distribution can be treated as static between two relevant time shot of the graph at $t$ and $t+1$. While this assumption may not strictly hold when processing batches, we consider that deriving the specific bounds will yield the same theoretical insights on the considered method's ability to propagate information.

## B  Proof of Theorem 2.2

**Theorem** (GCN-based aggregation). *Let's consider a CTDG-based function $f : (\mathcal{A}, \mathcal{X}) \to \mathcal{Y}$ based on $L$ GCN-like layers. After an event between node $i$ and another node, we have the following:*

- *For any node $u$ not involved in the event and for which $L < d(u, i)$ we have $u$ is $\gamma - flowing$ with:*

$$d_{\mathcal{Y}}(f_u(G_{t+1}), f_u(G_t)) \leq \hat{w}_u \|W_t\| \prod_{l=1}^{L} \|W^{(l)}\|.$$

- *For any node $u$ not involved in the event and for which $L \geq d(u, i)$ we have $u$ is $\gamma - flowing$ with:*

$$d_{\mathcal{Y}}(f_u(G_{t+1}), f_u(G_t)) \leq \prod_{l=1}^{L} \|W^{(l)}\| [\hat{w}_u \|W_t\| + \hat{w}_{u,i} \underset{t,t+1}{\Delta}(s_i)],$$

*with $\hat{w}_u$ being the sum of temporal normalized walks of length $(L-1)$ starting from node $u$ and $\hat{w}_{u,i}$ is the normalized shortest path between $u$ and $i$.*

*Proof.* We consider the case in which the propagation is done using a GCN-like aggregation. Specifically, we recall that a GCN update at layer $\ell$ for a node within a neighborhood can be written as:

$$h_u^{(\ell)} = \sigma^{(\ell)} \left( \sum_{v \in \mathcal{N}(u) \bigcup \{u\}} \frac{W^{(\ell)} h_v^{(\ell-1)}}{\sqrt{(1 + \deg(\text{u}))(1 + \deg(\text{v}))}} \right) \tag{5}$$

where $W^{(\ell)} \in \mathbb{R}^{e_{\ell-1} \times e_\ell}$ is the learnable weight matrix with $e_\ell$ being the embedding dimension of layer $\ell$ and $\sigma^{(\ell)}$ is the activation function of $\ell$-th layer.

**Proof's Intuition.** In our proof, we want to see the effect of an event on the node representation of a node $u$ within our graph. Typically, the set of nodes can be divided into two main categories. One first set which is within a reachable distance of where the event happened in respect to the number of message-passing layers. A second set, which is not reached based on the chosen number of message-passing. We approached the proof therefore by considered each set independently, where we upper-bound the expected change in each set. This latter division gives an intuition on how the considered model $f$ interacts with the different sets and consequently reflects its ability to propagate information within the graph after an event occurrence.

### B.1   For nodes not involved in the event and for which $L < d(u, i)$

In this part, we consider the nodes that are not directly connected in the events and that are distant from the nodes involved in the events. Specifically, we consider that the number of propagation $L < d(u, i)$ with $u$ being the node and $i$ being the node included in the event and $d$. In this proof, we consider that we are dealing with a GCN-like propagation, we can therefore right the following:

$$d_{\mathcal{Y}}(f(G^{t+1}), f(G^t)) = \|h_{u,t+1}^{(l)} - h_{u,t}^{(l)}\|$$

$$= \|\sigma^{(l)} \left( \sum_{v \in \mathcal{N}(u) \bigcup \{u\}} \frac{W^{(\ell)} h_{v,t+1}^{(\ell-1)}}{\sqrt{(1 + \deg(\text{u}))(1 + \deg(\text{v}))}} \right)$$

$$- \sigma^{(l)} \left( \sum_{v \in \mathcal{N}(u) \bigcup \{u\}} \frac{W^{(\ell)} h_{v,t}^{(\ell-1)}}{\sqrt{(1 + \deg(\text{u}))(1 + \deg(\text{v}))}} \right) \|$$

$$\leq \left\| \sum_{v \in \mathcal{N}(u) \bigcup \{u\}} \frac{W^{(\ell)} h_{v,t+1}^{(\ell-1)}}{\sqrt{(1 + \deg(\text{u}))(1 + \deg(\text{v}))}} - \sum_{v \in \mathcal{N}(u) \bigcup \{u\}} \frac{W^{(\ell)} h_{v,t}^{(\ell-1)}}{\sqrt{(1 + \deg(\text{u}))(1 + \deg(\text{v}))}} \right\|$$

$$= \|W^{(\ell)}\| \sum_{v \in \mathcal{N}(u) \bigcup \{u\}} \frac{\|h_{v,t+1}^{(\ell-1)} - h_{v,t}^{(\ell-1)}\|}{\sqrt{(1 + \deg(\text{u}))(1 + \deg(\text{v}))}}$$

We see that the right term on the resulting inequality is related to the one we are considering. Therefore, by iteratively doing the same process, we get the following:

$$d_{\mathcal{Y}}(f(G^{t+1}), f(G^t)) = \|h_{u,t+1}^{(l)} - h_{u,t}^{(l)}\|$$

$$\leq \prod_{l=1}^{l} \|W^{(l)}\| \sum_{z \in \mathcal{N}(y) \bigcup \{y\}} \frac{\|h_{v,t+1}^{(0)} - h_{v,t}^{(0)}\|}{\sqrt{(1 + \deg(u))}(1 + \deg(w))(1 + \deg(j)) \dots (1 + \deg(y))\sqrt{(1 + \deg(z))}}$$

$$\leq \prod_{l=1}^{L} \|W^{(l)}\| \sum_{z \in \mathcal{N}(y) \bigcup \{y\}} \frac{\|W_t\|}{\sqrt{(1 + \deg(u))}(1 + \deg(w))(1 + \deg(j)) \dots (1 + \deg(y))\sqrt{(1 + \deg(z))}}$$

$$\leq \hat{w}_u \|W_t\| \prod_{l=1}^{L} \|W^{(l)}\|$$

with $\hat{w}_u$ being the sum of temporal normalized walks of length $(L-1)$ starting from node $u$.

### B.2  For nodes not involved in the event and for which $L \geq d(u, i)$

Similar to the previous part, we have the result:

$$d_{\mathcal{Y}}(f(G^{t+1}), f(G^t)) = \|h_{u,t+1}^{(l)} - h_{u,t}^{(l)}\|$$

$$\leq \prod_{l=1}^{L} \|W^{(l)}\| \sum_{z \in \mathcal{N}(y) \bigcup \{y\}} \frac{\|h_{v,t+1}^{(0)} - h_{v,t}^{(0)}\|}{\sqrt{(1 + \deg(u))}(1 + \deg(w))(1 + \deg(j)) \dots (1 + \deg(y))\sqrt{(1 + \deg(z))}}$$

In the specific case that $L \geq d(u, i)$, we know that a part of the neighborhood contains one of the nodes that are subject to the event $i$. Therefore we have the following:

$$d_{\mathcal{Y}}(f(G_{t+1}), f(G_t)) \leq \prod_{l=1}^{L} \|W^{(l)}\| \sum_{z \in \mathcal{N}(y) \bigcup \{y\}} \frac{\|h_{v,t+1}^{(0)} - h_{v,t}^{(0)}\|}{\sqrt{(1 + \deg(u))}(1 + \deg(w))(1 + \deg(j)) \dots (1 + \deg(y))\sqrt{(1 + \deg(z))}}$$

$$\leq \prod_{l=1}^{L} \|W^{(l)}\| \left[ \sum_{z \in \mathcal{N}(y) \bigcup \{y\} \backslash \{i\}} \frac{\|h_{v,t+1}^{(0)} - h_{v,t}^{(0)}\|}{\sqrt{(1 + \deg(u))}(1 + \deg(w))(1 + \deg(j)) \dots (1 + \deg(y))\sqrt{(1 + \deg(z))}} \right.$$

$$\left. + \frac{\|h_{i,t+1}^{(0)} - h_{i,t}^{(0)}\|}{\sqrt{(1 + \deg(u))}(1 + \deg(w))(1 + \deg(j)) \dots (1 + \deg(y))\sqrt{(1 + \deg(i))}} \right]$$

The first term is like the previous approach only dependent on temporal projection aspect, while the second is rather dependent on the difference in terms of memory of the updated node $i$ involved in the event. We therefore find the following:

$$d_{\mathcal{Y}}(f(G_{t+1}), f(G_t)) \leq \prod_{l=1}^{L} \|W^{(l)}\| \sum_{z \in \mathcal{N}(y) \bigcup \{y\}} \frac{\|h_{v,t+1}^{(0)} - h_{v,t}^{(0)}\|}{\sqrt{(1 + \deg(u))(1 + \deg(w))(1 + \deg(j)) \dots (1 + \deg(y))}\sqrt{(1 + \deg(z))}}$$

$$\leq \prod_{l=1}^{L} \|W^{(l)}\| \big[\hat{w}_u \|W_t\| + \hat{w}_{u,i} \underset{t,t+1}{\Delta}(s_i)\big]$$

with $\hat{w}_u$ being the sum of temporal normalized walks of length $(L-1)$ starting from node $u$ and $\hat{w}_{u,i}$ is the normalized shortest path between $u$ and $i$.

$\square$

## C   Proof of Theorem 2.3

**Theorem** (Attention-based aggregation). *Let's consider a CTDG function $f : (\mathcal{A}, \mathcal{X}) \to \mathcal{Y}$ based on L attention-based layers. After an event between node $i$ and another node, we have the following:*

- For any node $u$ not involved in the event and for which $L < d(u,i)$ we have $u$ is $\gamma - \text{flowing}$ with:

$$d_{\mathcal{Y}}(f_u(G_{t+1}), f_u(G_t)) \leq deg(u)\big[\|W_t\| + B\|W_t\|^2\big].$$

- For any node $u$ not involved in the event and for which $L \geq d(u,i)$ we have $u$ is $\gamma - \text{flowing}$ with:

$$d_{\mathcal{Y}}(f_u(G_{t+1}), f_u(G_t)) \leq deg(u)\big[\|W_t\| + B\|W_t\|^2\big] + \Delta(s_i),$$

with $deg(u)$ being the degree of node $u$ and $B$ an upper-bound of hidden representation space.

*Proof.* **Proof's Intuition.** Similar to the previous proof, we consider in this case that the model is rather based on an attention framework to aggregate the information within the neighborhood. We start therefore by analyzing the effect of this addition. We afterwards, and in the same way as the GCN-case, derived the upper-bound for each set of nodes based on the distance to the event.

In this part we rather focus on attention-based aggregation within our temporal neighborhood, which can formulated as the following:

$$h_u^{(t+1)} = \sigma^{(\ell)} \left(\sum_{k \in \mathcal{N}(u) \bigcup \{u\}} \alpha_k^{(t)} h_k^{(t)}\right) \tag{6}$$

Let's consider a node $u$ at time $t$, we have the following:

$$\|h_u^{(t+1)} - h_u^{(t)}\| = \|\sum_{k \in \mathcal{N}_u(t+1)} \alpha_k^{(t+1)} h_k^{(t+1)} - \sum_{k \in \mathcal{N}_u(t)} \alpha_k^{(t)} h_k^{(t)}\|$$

$$\leq \sum_{k \in \mathcal{N}_u(t+1)} \|\alpha_k^{(t+1)} h_k^{(t+1)} + \alpha_k^{(t+1)} h_k^{(t)} - \alpha_k^{(t+1)} h_k^{(t)} - \alpha_k^{(t)} h_k^{(t)}\|$$

$$\leq \sum_{k \in \mathcal{N}_u(t+1)} \|\alpha_k^{(t+1)}[h_k^{(t+1)} - h_k^{(t)}] + h_k^{(t)}[\alpha_k^{(t+1)} - \alpha_k^{(t)}]\|$$

From the previous inequality, we see two main effects. A first effect which is dependent on the attention mechanism and a second effect which is dependent on the neighbor's representations.

We recall that the attention mechanism is formalized as follows:

$$\alpha_{ij}^{(t)} = \frac{\exp\left(e_{ij}^{(t)}\right)}{\sum_{k \in N(i)} \exp\left(e_{ik}^{(t)}\right)}, \tag{7}$$

with:

$$e_{ij}^{(t)} = \text{LeakyReLU}\left(w^\top [s_i^{(t)}, s_j^{(t)}]\right), \tag{8}$$

where $s_i^{(t)}$ is the "memory' of node $i$ at time $t$, $w$ is a learnable weight vector, and $[s_i^{(t)}, s_j^{(t)}]$ denotes the concatenation of $s_i^{(t)}$ and $s_j^{(t)}$. In this perspective, and since Leaky-ReLU is 1-Lipschitz continuous Virmaux & Scaman (2018), we have the following:

$$\|\alpha_k^{(t+1)} - \alpha_k^{(t)}\| \le \|w\|([s_i^{(t+1)}, s_i^{(t)}] - [s_u^{(t+1)}, s_u^{(t)}]) \tag{9}$$
$$\le \|w\|\delta(s_u)\delta(s_i) \tag{10}$$

### C.1 For any node $u$ not involved in the event and for which $L < d(u, i)$

Let's consider a node $u$ that is not involved in the event and for which $L < d(u, i)$. The elements within the neighborhood of this specific node are not subject to memory update. Additionally, for the node $u$, the difference in memory is only dependent on the time projection. Hence, if we assume that this is linear, we have the following:

$$\|\alpha_k^{(t+1)} - \alpha_k^{(t)}\| \le \|w\|\|W_t\|^2 \Delta_t^2$$

Without loss of generality, we additionally assume that the representation space is bounded; specifically for each node $u \in \mathcal{N}$ at any timestamp $t$, there exists a finite constant $B_u^{(t)} \ge 0$ such that:

$$\|h_u^{(t)}\| \le B_u^{(t)}, \quad \text{where } B = \max_{u \in \mathcal{N}, \, t} B_u^{(t)} \implies \|h_u^{(t)}\| \le B \text{ for all } u \text{ and } t$$

From these two elements, we find the following:

$$\|h_u^{(t+1)} - h_u^{(t)}\| \le \sum_{k \in \mathcal{N}(u)} \Delta(s_k) + B\|W_t\|^2 \Delta_t^2$$

### C.2 For any node $u$ not involved in the event and for which $L \ge d(u, i)$

Let's now consider a node $u$ that is not involved in the event and for which $L \ge d(u, i)$. For the considered node $u$, one of the nodes $i$ included in the event is within the neighborhood, then its memory will be updated as well. From the previously derived inequality, we can directly write:

$$\|h_u^{(t+1)} - h_u^{(t)}\| \le \sum_{k \in \mathcal{N}(u)} \Delta(s_k) + B\|W_t\|^2$$
$$\le \left[\sum_{k \in \mathcal{N}(u)\setminus\{i\}} \Delta(s_k) + B\|W_t\|^2\right] + \Delta(s_i) + B\|W_t\|^2$$
$$\le \sum_{k \in \mathcal{N}(u)\setminus\{i\}} \|W_t\| + B\|W_t\|^2 + \Delta(s_i) + B\|W_t\|^2$$
$$\le deg(u)\left[\|W_t\| + B\|W_t\|^2\right] + \Delta(s_i)$$

$\square$

## D   On the Tightness of the Provided Upper-Bound

In Theorem 2.3, we derive an upper bound on the expected norm difference in a node's representation between two consecutive time steps. To evaluate the tightness of this theoretical bound, we analyze the TGN model (Rossi et al., 2020b) since it is the closest model to align out with our proposed theoretical framework that is adapted on the general framework of a CTDG function presented in Section 2.1.

We note that in our theoretical proof, we don't consider event features (which are related to the added edges). Consequently, in order to align with our desire to study the tightness of the bounds, we make an adaptation of our derivations to take into account a TGN.

We adapt the Equation equation 7 to take into account edge features as it is the case in the TGN model, we have the following:

$$\alpha_{ij}^{(t)} = \frac{\exp\left(e_{ij}^{(t)}\right)}{\sum_{k \in N(i)} \exp\left(e_{ik}^{(t)}\right)}, \tag{11}$$

with:

$$e_{ij}^{(t)} = \text{LeakyReLU}\left(w^\top [s_i^{(t)}, s_j^{(t)}] + w_2^\top [e'_{i,j}]\right), \tag{12}$$

with $w_2$ being the learned weight matrix used with the edge features and $e'_{i,j} = [e_{i,j} \,||\, \Delta_t]$ is the concatenated vector of the edge features and the timestamp as explained in Rossi et al. (2020b). By following this formulation and using the triangular inequality, we can adapt the upper-bound provided in 9 as follows:

$$\|\alpha_k^{(t+1)} - \alpha_k^{(t)}\| \leq \|w\|\delta(s_u)\delta(s_i) + \|w_2\|\|[e'_{i,j}]\|$$
$$\leq \|w\|\delta(s_u)\delta(s_i) + \|w_2\|[\|e_{i,j}\|^2 + \Delta_t\|^2]$$

It is important to note that our theoretical analysis assumes the effect of a single event on the graph, meaning we consider one event per batch, therefore treating $t$ and $t+1$ as consecutive time steps and only one event coming in per time step. However, in real-world datasets, time is tracked in an absolute timeline, and events may not occur at evenly spaced intervals. To align with our theoretical framework, we normalize the time intervals in these datasets, effectively mapping the absolute timestamps to a relative scale that satisfies our assumptions.

Based on our previous adaptation, in Theorem 2.3, we derive an upper bound on the expected norm difference in a node's representation between two consecutive time steps. Specifically, we compute the difference in node representations using both our theoretical estimates and empirical values obtained from a trained model. Following the structure of the theorem, we examine two primary cases: **(1)** nodes within the neighborhood of nodes involved in events and **(2)** nodes outside this neighborhood with $d(u, i) > L$. Figure 2 illustrates the results in Log-Scale, highlighting the difference between the analytical upper bound and the empirical values observed when an event occurs.

We observe that the bounds are notably tighter in cases where $L \geq d(u, i)$, as the representation differences are primarily influenced by the temporal projection component, in line with our theoretical analysis. However, for nodes within the neighborhood of the event, the proposed upper bound is somewhat looser, likely due to simplifying assumptions made in the theoretical derivation.

## E   Empirical Insights of the Information-Flow

As discussed in Section 2.2, the concept of Information-Flow serves as a metric for assessing the influence of an event on the graph's node embeddings. In particular, when a new edge is added (i.e., an event occurs), it is expected that the node embeddings will be updated accordingly. If no update is observed, for example, in an edge prediction scenario, the corresponding probabilities remain unchanged, indicating that the event

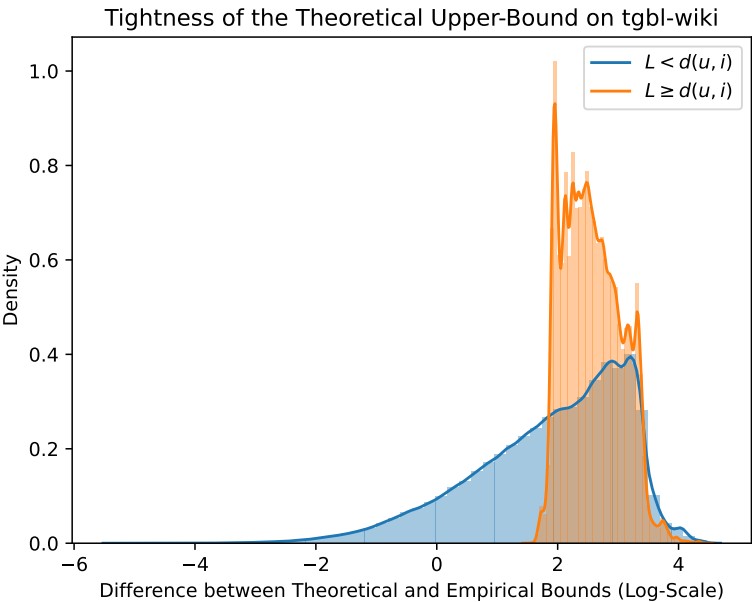

Figure 2: Difference between the theoretical upper-bound and the empirical values of the difference in norms for a node $u$'s representation.

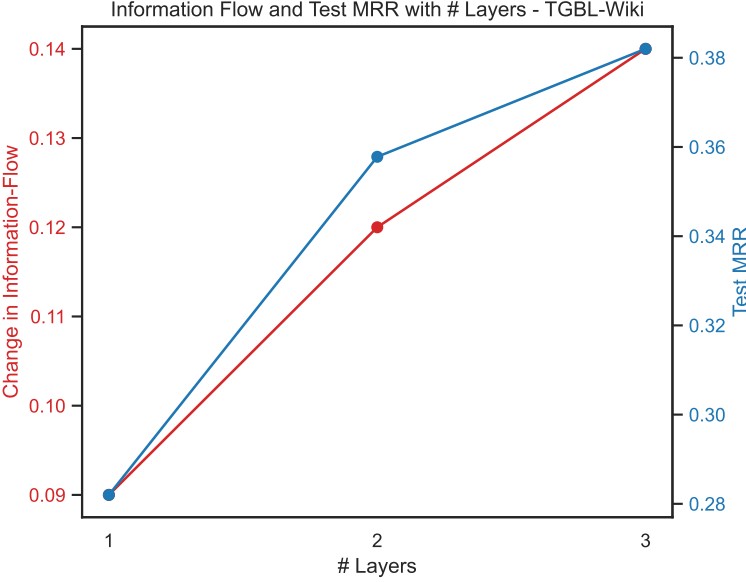

Figure 3: Information Flow change after an event occurring and the Average Test MRR on the TGBL-Wiki dataset for different number of layers.

provides no additional information. Ideally, the embeddings of all nodes in the graph should be updated in response to the event.

From the theoretical results, one key determinant of improved Information-Flow is the number of message-passing layers. To illustrate this, the normalized average norm difference of node embeddings is evaluated following an event in the TGBL-Wiki dataset, while varying the number of message-passing layers. Specifically, the TGAT model is considered with different configurations ($L = 1, 2, 3$) during inference. As depicted in Figure 3 (left axis), models with a larger number of layers exhibit a greater degree of information propagation across the graph.

However, it should be noted that although increased information flow is generally desirable, it does not necessarily lead to superior downstream performance. This discrepancy is often attributed to over-smoothing, a phenomenon well-documented in the literature. In certain graph topologies, such as bipartite structures, a model with a smaller number of layers can outperform deeper models. Figure 3 (right axis) further reports the Average Test MRR for the same TGAT models, highlighting this effect.

## F    Visual Summary of the Theoretical Insights

Our work position itself as a review of the available methods in the context of CTDGs. Specifically, our primary goal is to offer a unifying perspective on CTDG methods, especially in self-supervised settings, so that given a graph structure and constraints, the user can choose the right method to be used to maximize the performance. In this perspective, our provided "Information-Flow" in Section 2.2 studies how a method's components can affect its ability to propagate and update information after an event occurrence. This is relevant to understand how different methods shall interact with different topologies. In this perspective, Table 4 provides a visual summary of the different insights that were derived from our theoretical analysis and the computed upper bounds. Specifically, it highlights the key components influencing the "expressivity" of a temporal graph function, detailing their functional roles, associated mathematical constraints, and practical implications. The table summarize how factors such as memory usage, projection mechanisms, neighborhood depth, event features, and attention mechanisms contribute to the propagation of information over time, while also delineating the trade-offs in computational complexity and model stability.

| Component | Description | Bound Terms | Practical Effect |
|---|---|---|---|
| **Memory Usage** | Tracks historical interactions for each node. | $\Delta(s_i)$: Change in memory state for nodes involved in events. | Improves temporal dependency modeling; sensitive to memory size and update mechanism. |
| **Temporal Projection** | Projects node representations forward during inactivity. | $|W_t|$: Temporal projection weight matrix norm. | Captures time-varying node states but adds sensitivity to temporal sparsity. |
| **Aggregator Type** | Mechanism for combining neighborhood information (e.g., summation, attention). | Aggregator-dependent: $\prod_{l=1}^{L}|W^{(l)}|$ for GCN, degree-based terms for attention. | Influences expressivity and computational cost; enables fine-grained updates. |
| **Neighborhood Depth** ($L$) | Number of layers in the message-passing scheme, affecting the propagation range. | $\hat{w}_u, \hat{w}_j$: Normalized walks or shortest paths. | Controls the range of information flow but risks over-smoothing or over-squashing. |
| **Event Features** | Includes edge-related attributes (e.g., interaction type, timestamps). | $|W_t| + \mathcal{B}$: Interaction with edge features through projection and event-related adjustments. | Enhances representation with context-specific interactions but may increase model complexity. |
| **Attention Mechanism** | Dynamically weighs neighbor contributions using learnable parameters. | Degree-based terms $(deg(u))$ and learned weights $(W)$. | Enables selective information propagation, particularly useful for graphs with heterogeneous links. |
| **Graph Topology** | Structural characteristics such as sparsity, degree distribution, and community structures. | $\hat{w}_u, \hat{w}_j$: Impact of topology on walk-based terms. | Affects long-range dependency modeling and suitability for specific graph types (e.g., bipartite). |

Table 4: Visual Representation of the theoretical insights from the computer upper-bounds in Theorem 2.3 and 2.2.

Table 5: Parameters for generating the stochastic block model dataset.

| B | N | $N_B$ | $p_{i,j}$ | $p_{i,i}$ | $t_{gen,kc}$ |
|---|---|---|---|---|---|
| 100 | 1000 | 30 | 0.025 | 0.25 | $[6, 20)$ |

## G  Datasets and Implementation Details

### G.1  Synthetic Datasets

To perform further experimentation and to validate our theoretical insights, we evaluated our considered benchmarks on a number of graph types. Since finding accessible real datasets has been a challenge, we have opted to rather use synthetic and generated datasets in which we can control the different settings and confirm our hypothesis. In the following section, we describe in details our data generation process and we give additional information to easily reproduce this process.

**Long-Range Benchmark**  Long-range graphs require long-range reasoning to achieve high performance, as interactions between distant nodes or graph regions can significantly influence outcomes. To effectively capture these relationships, specific information propagation strategies are necessary, as standard message-passing techniques often fall short in handling these long-range dependencies.

Our objective is to evaluate the performance of the reviewed benchmarks in handling this category of graphs. Following the methodology from prior work (Gravina et al., 2024), we introduce a temporal component to the static PascalVOC-SP graph from the Long-range benchmark dataset (Dwivedi et al., 2022). This dataset comprises graphs generated from images, where each node represents a region within the image that belongs to a particular class. The temporal dimension is introduced by assuming an ordered sequence in node appearance, progressing from the top-left to the bottom-right of the image. For further details, we refer readers to the original work (Gravina et al., 2024).

**Community-based Graphs.**  One part of our experimental evaluation consists of considering a graph which is based on communities so as to assess the method's ability to propagate information within different clusters. In this perspective, we first generate $B$ blocks of dense communities with $N_B$ nodes in each community, resulting in $N$ nodes in separate disconnected components with a connection density of $p_{i,i} \forall$. At each timestep $t$, we select $k_c = 4$ pairs of communities that we (pairwise) connect over a random number $t_{gen,k_c} \in [6, 20]$. Specifically, this selection follows a normalized density of $\frac{p}{10} = 0.025$, normalized over a sampled time horizon $t_{gen,k_c}$ over which we connect the pair of clusters. Naturally, as time progresses, the graph becomes progressively more connected, and its topology evolves accordingly. The used parameters are reported in Table 3. For the evaluation, we leverage the normalized sampling densities between cluster pairs, derived from the data generation procedure, to identify which communities are most likely to be connected. Based on this, we formulate the task of ranking nodes within the active clusters and evaluate the results using Normalized Discounted Cumulative Gain (NDCG) (Järvelin & Kekäläinen, 2002). We choose the NDCG here, as we know which communities should be more and less likely to be connected during that specific timestep and their *true* relevance score as they are directly derived from the sampling of stochastic block model used.

**Bi-Partite Datasets.** For this category of graphs, we have focused on using the real world graph TGBL-Wiki dataset, which is part of the TGB benchmark datasets (Huang et al., 2024). The dataset is a bi-partite network where Wiki pages and editors are nodes, while an edge is added when a user edits a page at a specific timestamp.

**Homogeneous Graphs.**  We additionally experimented with homogeneous graphs, i.e. graphs where nodes and edges belong to the same type. This type of graph encompasses a wide range of applications, including social networks, transportation systems, and other domains where graph structures are uniform. For this evaluation, we used an adaptation of the real world TGBL-Coin dataset which is part of the TGB benchmarks (Huang et al., 2024) too. This dataset contains cryptocurrency transactions, where each node

is an address and each edge denotes the transfer of funds from one address to another at a certain time. The results are reported in Table 7.

For both the real and synthetic datasets, characteristics and information about the graphs utilized in the experimental results of the study are presented in Table 6.

Table 6: Statistics of the graph datasets used in our experiments.

| Dataset | #Nodes | #Edges | #Edge Features | #Surprise Index |
|---|---|---|---|---|
| T-PascalVOC 10 | 2,671,704 | 2,660,352 | 14 | 1.0 |
| T-PascalVOC 30 | 2,990,466 | 2,906,113 | 14 | 1.0 |
| TGBL-Wiki | 9,227 | 157,474 | 172 | 0.108 |
| SBM Graph | 3,000 | 172,570 | 0 | 1.0 |

Table 7: Average Test and Validation MRR ($\pm$ denotes standard deviation) of the different approaches on an adaptation of the real-world dataset TGBL-Coin. Best performance in **bold**.

| Model | Homogeneous Graph | |
|---|---|---|
| | Test | Val |
| JODIE | $0.713 \pm 0.028$ | $0.552 \pm 0.131$ |
| DyRep | $0.688 \pm 0.007$ | $0.697 \pm 0.007$ |
| TGN | $0.738 \pm 0.007$ | $0.731 \pm 0.006$ |
| TGAT | $\mathbf{0.748 \pm 0.025}$ | $\mathbf{0.765 \pm 0.023}$ |
| CTAN | $0.709 \pm 0.003$ | $0.754 \pm 0.006$ |

## G.2 Experimental Details

For our experimental setting on the **long-range datasets**, we used the Adam optimizer (Kingma & Ba, 2015) with a 1e-04 learning rate and a batch size of 256 for training, validation and testing. We set the patience to 20 and we optimize the F1-Score, and the learning consisted of 20 epochs. All the experiments were run 5 times to reduce the effect of initialization and other randomization parameters. For the model's parameters, we directly used the best parameters reported in previous work (Dwivedi et al., 2022).

For our experimental setting on the real-world **bi-partite dataset** TGBL-Wiki, we used the Adam optimizer with a 1e-04 learning rate and a batch size of 200 for training, validation and testing. We set the patience to 20 and we optimize the Mean Reciprocal Rank (MRR) score, and the learning consisted of 50 epochs. All the experiments were run 5 times to reduce the effect of initialization and other randomization parameters. The number of GNN layers is $L = 1$, while the number of heads (for multi-head attention) is 1 (for CTAN) or 2 (for TGAT and TGN). For CTAN, epsilon is 1.0 and gamma is 0.1, as reported in Gravina et al. (2024). The rest of the configuration is taken from the TGB codebase: the weight decay penalty is 0, the embedding/time/memory dimension is 100, the neighborhood sampler size is 10, and evaluation is done with the TGB evaluator.

Finally, for our experiments on the **community-based** SBM dataset and **homogeneous** adapted TGBL-Coin dataset, we have set the number of layers to $L = 1$ and a unique attention head. Similarly to the previous experiment, we have used the Adam optimizer (Kingma & Ba, 2015) with a 1e-04 learning rate and a batch size of 200. All the models were trained for 30 epochs with the patience set to 5, and we have used a hidden dimension of 100. All the experiments were run 5 times to reduce the effect of initialization and other randomization parameters. For CTAN, epsilon is 1.0 and gamma is 0.1, as reported in Gravina et al. (2024).

The source code necessary to reproduce our experiments is publicly available at: `https://github.com/king/ctdg-info-flow`

