# OpenReview forum: "Expressivity of Representation Learning on Continuous-Time Dynamic Graphs: An Information-Flow Centric Review"
_TMLR — Accepted by TMLR_

### Review · Reviewer_s8Qu · 2024-12-20

**Summary Of Contributions:**

This paper reviews Graph Representation Learning (GRL) on Continuous-Time Dynamic Graphs (CTDGs) with a focus on Self-Supervised Representation Learning (SSRL). It presents an innovative Information-Flow (IF) centric theoretical framework to assess expressivity in CTDG models, categorizes existing methods based on their applicability to various graph types, and supports theoretical findings with empirical evaluations.

**Audience:**

Yes

**Claims And Evidence:**

Yes

**Requested Changes:**

1. The review around contrastive pretraining needs a further polish to include more recent work and how those methods relate to the proposed framework.

2. Please discuss the limitations of key assumptions in the theoretical framework and their implications for practical applications.

3. Minor Corrections:
 - In a few places “None-MP” should be “Non-MP”.
 - The title of 3.2 should be “Self-Supervised Representation Learning (SSRL)”

**Strengths And Weaknesses:**

**Strengths**

- The paper is well-organized, with a clear progression from theoretical foundations to practical evaluations, making it easy to read.

- The proposed IF-centric framework introduces a new way to analyze expressivity in CTDGs, potentially enhancing understanding of temporal and structural dynamics.

- By highlighting SSRL, the paper addresses existing CTDG literature gaps and underscores the potential for learning robust representations without labeled data.

- The experimental evaluations support the theoretical claims and offer insights into method selection for specific tasks.

**Weakness**

- The theoretical framework assumes specific properties (e.g., 1-Lipschitz functions, stable degree distributions) that may not always hold in real-world scenarios. The authors may provide more justification on this.

- The discussion on contrastive pretraining is valuable but lacks references to works such as [1] and [2].

- The synthetic datasets are diverse, but there's limited experimentation with real-world datasets. However, as a survey paper, this might not be necessary.

[1] Xu, Yiming, et al. "CLDG: Contrastive learning on dynamic graphs." 2023 IEEE 39th International Conference on Data Engineering (ICDE). IEEE, 2023.

[2] Zhu, Zulun, et al. "Topology-monitorable Contrastive Learning on Dynamic Graphs." Proceedings of the 30th ACM SIGKDD Conference on Knowledge Discovery and Data Mining. 2024

---

> ### Author Response · Authors · 2025-01-07
> **Response to Reviewer Comments and Revisions Summary**
>
> We thank the reviewer for the insightful comments and for spotting the minor corrections. We have edited accordingly. In what follows, we respond to some of the questions that the reviewer has introduced:
>
> **Regarding the theoretical assumptions**
> We note that the considered theoretical assumptions are very realistic. Specifically, the majority of used activation functions are actually 1-Lipschitz. Additionally, since we consider the effect of a single event, assuming the node’s distribution to be similar is very realistic. We agree that a thorough discussion of these assumptions was missing from the manuscript and we have added it in the Appendix A.
>
> **Regarding contrastive pretraining**
> We thank the reviewer for the references and we apologize for missing these references. We have updated the section to include the approaches.
>
> **Regarding the experiments**
> We have mainly focused on the synthetic dataset in our first draft given that we can control the graph dynamics. The main goal was to show that our proposed categorization is valid empirically. Based on your proposition, we have also added additional results (cf. Appendix F and Table 6) on a variation of the TGBL-Coin real world dataset, therefore covering multiple types of graphs.

---

> > ### Comment · Reviewer_s8Qu · 2025-01-13
> > **Thank you for your response.**
> >
> > I thank the authors have resolved my concerns.

---

### Review · Reviewer_TCWK · 2024-12-24

**Summary Of Contributions:**

This work delivers a review of representation learning on dynamic graphs (specifically CTDGs), with a focus on SSRL. The authors derived a theoretical framework to analyze the expressivity upper-bound of different existing models, categorizes them according to a taxonomy related to the framework. They also conducted empirical validations.

**Audience:**

Yes

**Claims And Evidence:**

Yes

**Requested Changes:**

The analysis of the theoretical upper-bound versus empirical results should be moved from the appendix to the main paper. This segment deserves more prominence, as I mentioned already. Expanding this analysis with results from additional datasets would further strengthen the paper’s empirical contributions.

Can authors give out their judgement in a promising category of methods (or component of methods) using the expressivity framework?
Please make clear of the availability plan of the (synthetic) datasets and code.

Lastly, I noticed the way Theorem 2.2 and 2.3 are expressed is different in format, I think it is the best to make them identical and please try to make the proof in Appendix easier to read for general audiences – probably add a bit more explanation?

**Strengths And Weaknesses:**

Strengths:

1. The information flow centric theoretical framework is a new ground in understanding the expressivity of CTDG models. It focuses on quantifying the ability of models to encode and propagate temporal and structural information.

2. The categorization of existing methods according to their compatibility with different graph structures and temporal dynamics is informative. It provides another view of the complex landscape of CTDG methods, potentially making it easier for practitioners to select or adapt approaches for their specific needs.

3. The paper is generally written well, with a visible effort of connecting its theoretical contributions with practical insights. The empirical validation on long-range, bi-partite and community-based graphs provides an initial insight in understanding the relative capabilities of the tested methods .

Weaknesses:

Some aspects in the paper could benefit from additional elaboration or a broader perspective. One area that stood out is the analysis of the difference between the theoretical upper-bound and the empirical values of node representation norms, which only present thinly in the appendix. I believe this kind of analysis is not only important but also essential for appreciating the practical utility of the theoretical framework.

Minor: lack of information on the availability of datasets and code.

---

> ### Author Response · Authors · 2025-01-07
> **Response to Reviewer Feedback and Planned Revisions**
>
> We would like to sincerely thank the reviewer for their feedback and comments.
>
> **Regarding the empirical tightness of the upper-bounds**
> We agree with the reviewer’s assessment about the necessity of validating empirically the tightness of the upper-bound and hence understanding the utility of the proposed “Information-Flow” quantity. Nonetheless, the aim of our work is rather to provide a thorough review of the available methods tackling CTDGs in a SSL manner. In this perspective, we have chosen to place the empirical validation of the bounds in the Appendix to remain within the proposed 12-pages limits. We will therefore move this section, with additional datasets to the main paper in the final version.
>
> **Promising category of methods**
> From the theoretical perspective, it seems that the main elements that control the information propagation are the temporal encoding and the message-passing framework. We additionally see that the main drawback  of the actual method is the ability to propagate information after an event in nodes that are not directly within the k-hop neighborhood. In the same perspective, approaching the problem through methods of connectivity (such as dynamical rewiring) is very complex and time consuming. In this direction, we find that an interesting approach is to connect the different nodes within different clusters/parts promoting both local and global connectivity within the graph.
>
> **Availability plan of the (synthetic) datasets and code**
> We plan to make the code for generating the datasets and reproducing the experiments publicly available on GitHub, which we will disclose in the camera-ready version.
>
> **Regarding the theorems and the proofs**
> We thank the reviewer for pointing out this element, we have now make the Theorems in the same format, please see the changes in Appendix B and C. If there are further recommended changes concerning Theorem 2.2 and 2.3, please let us know.

---

> > ### Author Response · Authors · 2025-02-26
> > **Update about the code availability**
> >
> > Dear Reviewer,
> >
> > Thank you again for your valuable feedback and suggestions. We would like to point out that our code is now provided in the Supplementary Material section and will be made publicly available on GitHub after publication.
> >
> > We hope we have addressed all your concerns and questions. We are also happy to further investigate any remaining points if needed.

---

### Review · Reviewer_8yma · 2024-12-29

**Summary Of Contributions:**

The authors offer a survey of the field of Continuous-Time Dynamic Graphs (CTDG), organized in a taxonomy of Message Passing (MP), non-MP, and hybrid methods. To illustrate the expressiveness of the methods, the authors study a general framework incorporating many (but not all) of the listed models, from the point of view of Information Flow (IF). In particular, the authors's study focuses on upper bounding the flow quantity, defined as the distance between embeddings before and after an event in the graph. Finally, a section examines the use of Self Supervised Learning for CTDG downstream tasks. The authors also include an experimental section for benchmarking some of the surveyed methods.

**Audience:**

Yes

**Claims And Evidence:**

No

**Requested Changes:**

Critical changes:

1 - more in-depth discussion of theoretical results, with better motivations in claims over how the flow quantity relates to expressivity.

2 - more in-depth application of the theory on the surveyed methods.

3 - clearer empirical proofs regarding how the theory drives better results, for example, pinpointing how the upper bound correlates with performance, depending on the architecture.

4 - explaining clearly the role of the paper, and how self-supervised learning is useful to the narrative.

**Strengths And Weaknesses:**

Strengths:

1 - the paper seems to be typo-free and has good English. It is generally well-written.

2 - the survey is thorough in the list of methods

3 - the Information Flow point of view is interesting in its principles

Weaknesses:

1 - I think I've missed the point of the paper: on the one hand, this paper is a survey that categorizes many methods, but on the other, it also introduces some theoretical concepts like IF. Although this can work in principle, as IF is used to address the expressivity of methods, it is also restricted to a limited number of methods, namely those with Message-Passing layers (like Graph Conv. Networks and Attention-based).

2 - I did not understand the relation between the flow quantity (and its upper bound) and the expressivity of a model. Also, I think the application of theorems 2.1 and 2.2 is not discussed in depth enough for each surveyed method. A visual explanation might also help, for example, a plot #layers L vs. upper bound in an example network.

3 - the introduction of Self Supervised Learning seems out of place in the context of the paper, as there is no mention of linking it with the theory developed in Section 2.2 nor in the Experiment section.

4 - I think the experimental section fails in its premise, that is, to prove the theoretical insights of the paper. The results show that CTAN, a state-of-the-art method, surpasses all other methods. This, in my view, is stating the obvious, as it is, in fact, a very recent state-of-the-art method. This does not validate the theoretical results, and no clear correlation is shown.

5 - Although the paper is generally well-written, it misses clarity in some of its statements, for example: "However, we argue that JODIE struggles with stochastic events, especially in multi-community graphs, where sudden, unpredictable changes are more
prevalent.". Do you have proof for this claim?

---

> ### Author Response · Authors · 2025-01-09
> **Part 1 of 3: Clarifying the paper’s aim, scope, and linking IF to model expressivity**
>
> We would like to sincerely thank the reviewer for carefully reading our work and providing constructive feedback. Below, we address each concern in detail.
>
> ## Clarity on the paper’s aim and scope
>
> We acknowledge the reviewer’s comment that our paper appears to serve multiple roles: (1) a survey of existing CTDG methods, (2) an introduction of the Information-Flow (IF) theoretical framework, and (3) a discussion of Self-Supervised Representation Learning (SSRL) in CTDGs. Our primary goal is to offer readers a unifying perspective on CTDG methods, especially in self-supervised settings, so that they can choose methods that best suit the structure and constraints of their own graphs.
>
> To achieve this, we introduced IF to illustrate how specific components (e.g., GCN-like versus attention-based aggregations, temporal memory, etc.) affect the global ability of the model to propagate new information throughout the graph. This theoretical viewpoint then underpins our categorization of methods. Finally, because labeling is often expensive in real-world dynamic graphs, we discuss SSRL tasks (predictive and contrastive) as increasingly vital tools in this space.
>
> We clarified this overarching motivation in both the Introduction and Conclusion, emphasizing that the IF analysis is intentionally modular (i.e., it focuses on a general “CTDG function”) and downstream-task agnostic, making it suitable for a variety of learning paradigms, including SSRL.
>
> ## Linking IF to "expressivity"
>
> We appreciate the reviewer highlighting that the link between IF bounds and model expressivity was not sufficiently clarified. Below is our intended rationale:
>
> - **Conceptual definition**: In many graph representation contexts, “expressivity” refers to whether a GNN (or variant) can distinguish different substructures, often formalized via WL-like tests. In dynamic scenarios, the notion of “expressivity” needs to account for how well the model assimilates each new event and propagates this information to relevant (even distant) parts of the graph.
>
> - **Our IF metric**: The IF lens measures how much a given event (e.g., adding an edge) changes node embeddings globally. If an event yields no change in embeddings for most nodes, then that event has not “flowed” anywhere in the graph, suggesting the model may fail to incorporate important signals from that event (e.g., user-item interaction). Conversely, if new events lead to more widespread or deeper updates, the model is arguably more expressive of the newly arrived information.
>
> - **Practical implications**: A model with too few layers (or missing memory and temporal projection) can have low IF for distant nodes, meaning it struggles to incorporate new signals (specially in long-range dependencies datasets). By contrast, a multi-layer model with attention or diffusion-based propagation tends to have higher IF when needed, thus capturing more of the graph’s evolving structure.
>
> To make this connection clearer, we have added a preliminary analysis (Appendix F in the updated draft) on the TGBL-Wiki dataset where we investigate the effect of the number of layers on the information-flow quantity and also on the performance on the downstream task. Specifically, we can see that as expected, increasing the number of layers results in a better information propagation within the graph. We will provide additional results on other datasets in the next paper’s draft.

---

> > ### Comment · Reviewer_8yma · 2025-01-10
> >
> > Dear authors,
> > thank you for thoroughly examining my review, and for the explanations provided.
> >
> > Thanks for clarifying the scope of the paper. About the expressivity, reading your explanation (and re-reading the manuscript) I came to understand the bound on IF as a sort of "capacity" of the model, which can (or won't if not needed) change the embeddings up to the bound. Now, what still troubles me is the dependency on the scale of the embedding space and weights: in principle, a shallow GCN with bigger weights (in magnitude) could have a higher IF bound than a deeper GCN with smaller weights. Correct me if I'm wrong on this point.
> >
> > Another problem is over-smoothing, where having too many layers of message passing can lead to homogeneous representations, which in turn causes worse performance. This is not considered in IF, which seems to suggest that a higher number of layers leads to a higher bound. In particular, IF doesn't take into account the expressivity from the WL-test point of view. I think there should be a clear distinction to be made in the paper. I'm open to discussing more on this point.
> >
> > Finally, I can't find the preliminary analysis you mentioned in Appendix F of the revised manuscript, just the datasets discussion and experimental details.

---

> ### Author Response · Authors · 2025-01-09
> **Part 2 of 3: More thorough application of the IF theory to the surveyed metho**
>
> ## More thorough application of the IF theory to the surveyed methods
>
> While we aimed to keep the theoretical exposition and the method descriptions moderately separate (so that the survey remains reader-friendly), we understand that highlighting how each method either does or does not maximize IF can be useful.
>
> - **High-level strategy**: We will insert short "theory application" callouts in the beginning of Section 3 (Method Categorization and Review) summarizing how a method’s architecture (e.g., memory usage, layer design) relates to the bounds in Theorems 2.1 and 2.2. For instance, a method that relies heavily on single-hop updates with minimal memory might have a smaller upper bound on its IF for distant nodes, whereas a method with multi-layer attention and memory could potentially have a larger bound on node-embedding changes.
>
> - **Visual summary**: Where feasible, we will add a table/diagram matching key method components (e.g., memory usage, type of aggregator) with the relevant bound terms (e.g., presence of $\Delta(s_i)$, effect of $||W_t||$, etc.). This should make the connection between theory and practice more explicit. Here could be a starting point:
>
> | **Component**           | **Description**                                                                                         | **Bound Terms**                                                                                          | **Practical Effect**                                                                                 |
> |--------------------------|---------------------------------------------------------------------------------------------------------|----------------------------------------------------------------------------------------------------------|------------------------------------------------------------------------------------------------------|
> | **Memory Usage**         | Tracks historical interactions for each node.                                                           | $\Delta(s_i)$: Change in memory state for nodes involved in events.                                     | Improves temporal dependency modeling; sensitive to memory size and update mechanism.              |
> | **Temporal Projection**  | Projects node representations forward during inactivity.                                                | $\|W_t\|$: Temporal projection weight matrix norm.                                                     | Captures time-varying node states but adds sensitivity to temporal sparsity.                       |
> | **Aggregator Type**      | Mechanism for combining neighborhood information (e.g., summation, attention).                          | Aggregator-dependent: $\prod_{l=1}^L \|W^{(l)}\|$ for GCN, degree-based terms for attention.           | Influences expressivity and computational cost; attention enables fine-grained updates.            |
> | **Neighborhood Depth ($L$)** | Number of layers in the message-passing scheme, affecting the propagation range.                         | $\hat{w}_u$, $\hat{w}_{u, i}$: Normalized walks or shortest paths.                                   | Controls the range of information flow but risks over-smoothing or over-squashing.                |
> | **Event Features**       | Includes edge-related attributes (e.g., interaction type, timestamps).                                  | $\|W_t\| + B$: Interaction with edge features through projection and event-related adjustments.       | Enhances representation with context-specific interactions but may increase model complexity.      |
> | **Attention Mechanism**  | Dynamically weighs neighbor contributions using learnable parameters.                                   | Degree-based terms ($deg(u)$) and learned weights ($W$).                                             | Enables selective information propagation, particularly useful for graphs with heterogeneous links. |
> | **Graph Topology**       | Structural characteristics such as sparsity, degree distribution, and community structures.              | $\hat{w}_u$, $\hat{w}_{u, i}$: Impact of topology on walk-based terms.                              | Affects long-range dependency modeling and suitability for specific graph types (e.g., bipartite). |

---

> > ### Comment · Reviewer_8yma · 2025-01-10
> >
> > Thanks for the more structured explanation. I'm satisfied with this new integrated explanation. I feel it can make the reading more organic.

---

> ### Author Response · Authors · 2025-01-09
> **Part 3 of 3: The link/relation between IF framework and experiments/SSRL and more ...**
>
> ## Relating the experimental part to the theoretical insights
>
> The main goal of our theoretical analysis is to analyze the effect of each component within a CTDG function on the information propagation when an event occurs. We have therefore provided an upper-bound for a general CTDG function, which has provided us with a nomenclature to categorize the different methods in our review part. We therefore consider that our theoretical part has provided its value for our current work which is mainly focusing on surveying the methods. Typically, our work’s aim is to provide the reader with a clear categorization that could directly point out which method to use depending on which constraints/topology of the input graph.
>
> In our initial experimental results, we aimed to confirm two main points: (1) The tightness of the provided upper-bound (Appendix E), where we analyze how tight is the theoretical bound in respect to the empirical one. (2) An analysis on the performance of the surveyed methods on different graph topologies. While this latter doesn’t necessarily relate to our introduced IF, it does indeed confirm the theoretical insights that were derived (such as the expected performance of some methods in some topologies).
>
> Nonetheless, we agree with the reviewer’s assessment that the provided IF quantity can actually serve as a method to compare different methods in terms of "expressivity" by computing the specific upper-bound (both theoretically and empirically). In this direction, we have provided an initial investigation in which we consider the effect of the number of layers on the Information-Flow quantity and also the downstream performance in the updated draft (Appendix F). We consider therefore that the introduced concept of IF can help in future work to provide theoretical claims (in terms of performance) about a proposed method in comparison to the benchmarks.
>
>
> ## Integration of SSRL with the IF framework
>
> Our goal is to highlight how SSRL tasks (e.g., predictive or contrastive tasks) stand to benefit from expressive dynamic models. Large-scale dynamic graphs (recommender systems, social networks) often have expensive or sparse labels. Models that effectively propagate new signals (high IF) can offer more robust node or edge representations for these unlabeled scenarios. We will clarify this connection by adding the following to Section 3.2:
>
> - A statement explicitly noting that the same "event-based embedding changes" described in IF analysis also govern how well the model can handle self-supervised tasks that rely on detecting or contrasting new events.
>
> - Some add-on descriptions to explicitly point out how a high IF model is apt to excel in SSRL tasks that rely on subtle changes in node/edge states (e.g., contrastive tasks that need to discriminate small changes in a node’s neighborhood across time).
>
> While our main theoretical results do not directly mention SSRL loss functions, the capacity to propagate information strongly influences SSRL performance (e.g., predicting the next edge or effectively constructing negative samples in long-range neighborhoods). We will add these bridging contents to clarify the synergy between IF-based expressivity and SSRL.
>
> ## Other clarifications / claims
>
> Regarding the statement "JODIE struggles with stochastic events, especially in multi-community graphs...," we recognize the need for a more precise explanation or reference. Practically, JODIE’s updates are triggered largely by local events (i.e., user-item pairs), and it does not incorporate multi-hop message passing. As a result, if *unpredictable changes* occur in other communities far from this user-item pair, JODIE’s embeddings may not integrate that information quickly. Therefore, we (i) rephrase this claim more cautiously and (ii) reference the results in Table 3 (SBM dataset) that indeed show JODIE lagging behind multi-hop approaches when multiple communities become active.

---

> > ### Comment · Reviewer_8yma · 2025-01-10
> >
> > I think that better pinpointing the relation between IF and the methods of the survey (as proposed in your comment part 2) will make the link between IF and the experiments clearer. Again, I can't find the analysis you mentioned. Could you point me to it?
> >
> > Thank you for the additional explanations on SSRL. About the JODIE claim, I don't see the change in the revised version, nor do I see the fact that JODIE is lagging behind multi-hop methods in the SBM dataset (JODIE has 0.8491 test NDGC, which is between TGAT and TGN).

---

> ### Author Response · Authors · 2025-01-10
> **Further clarifications upon questions from reviewer 8yma**
>
> We thank the reviewer 8yma for his/her further reply and encouraging feedback.
>
> Indeed as framed by the reviewer, we approach the IF quantity from an upper-bound, hence analyzing the possibility (or not) of the changes in the embeddings. This is mainly due to the fact that deriving the quantity in a precise manner from a theoretical perspective is challenging (even impossible without some strong theoretical assumptions which we are trying to avoid to be realistic). In this perspective, having a bigger upper-bound can (with strong probability) reflect a big change in the node embeddings which is our main quantity to evaluate. This is indeed very different from how WL-like tests approach a graph function’s expressivity - and also understanding the over-smoothing /bottleneck problem that can arise in GNNs.
>
> As identified by the reviewer, three quantities are controlling the upper-bound (for instance in the case of Theo 1): (1) The weight of the message-passing, (2) The weight of the temporal encoder and (3) the topology of the graph (in terms of connectivity - normalized walks). Regarding the statement on a shallow GCN with bigger weights versus a deeper GCN with smaller weights, one should also take into account the topology component. In fact, this latter quantity depends on the walks of length $L-1$, with $L$ being the number of GCN layers. Hence, depending on the "connectivity" of the graph, the statement is valid or not but in general, it is indeed true. This latter point also relates to the question of over-smoothing as pointed out by the reviewer, where having a larger number of layers shouldn’t necessarily reflect a better IF.  We have tried to motivate this distinction in the paper (Sec 2.2), we apologize if this wasn’t enough and we will try to add more to that discussion.
>
> Thanks for pointing out the issue with the PDF. It seems there was a problem with our last manuscript update, apologies for that :) We have re-uploaded the PDF and double-checked that the correct version appears when using the generic PDF link in OpenReview (at the top of the page). If the issue persists on reviewer's end, we recommend navigating to the "Revisions" tab and retrieving the PDF from the latest revision record.

---

> > ### Comment · Reviewer_8yma · 2025-01-12
> >
> > Dear authors,
> >
> > thanks for the additional discussion and for clarifying further. I think the difference between expressivity under WL and IF really needs to be discussed a bit more in the main body of the paper, as we can't exclude that part of the better experimental performance is due to the architecture itself. For example, would you agree that CTAN, being based on continuous time GNNs, can have a high number of MPs, increasing the IF bound without incurring so much into the over-smoothing problem[1], whereas architectures like TGN and TGAT still suffer from the problem, not allowing to have a high IF bound?
> >
> > I can confirm that the issue with the PDF has been solved! On a stylistic note, I would consider merging figures 3 and 4 (having two vertical axes), which would compliment the comparison of IF vs test MRR. On the JODIE statement: I still don't see JODIE lagging behind so much in Table 3, at least not as much as in Table 1. On the opposite, it is performing better than TGAT. Wouldn't using Table 1 to prove the statement be stronger? Like before, I'm open for discussion.
> >
> >
> > [1] Xhonneux, Louis-Pascal, Meng Qu, and Jian Tang. "Continuous graph neural networks." In International conference on machine learning, pp. 10432-10441. PMLR, 2020.

---

> ### Author Response · Authors · 2025-01-13
> **Continued discussion with reviewer 8yma**
>
> We thank the reviewer for the discussion and for the further very interesting discussion. We have edited Figure 3 (Appendix F) accordingly and it indeed shows better the desired trade-off.
>
> We agree with the reviewer’s assessment, and we have edited Section 2.2 (the second paragraph)  accordingly to reflect more how we approach Expressivity based on our introduced IF and the WL-based tests. We thank the reviewer for pointing this out.
>
> We additionally share the reviewer’s point of view regarding CTAN and its superior performance due to the continuous time GNNs. Specifically, from our theoretical analysis, it seems that there exists a trade-off between the graph’s topology (in terms of connectivity) and the number of used layers. Hence, while stacking a higher number of layers (to reach long-dependencies nodes) can enhance the IF in the case of TGAT and TGN, it results in the effect of over-smoothing and therefore a loss in terms of downstream performance. Consequently, finding the right trade-off between the number of layers to propagate the information to all the graphs while avoiding over-smoothing is crucial and sometimes impossible for some topologies. In the other case of CTAN, increasing the number of layers doesn’t necessarily produce an over-smoothing effect and therefore can result in a better IF while having a good downstream task - which was seen in the experimental results. We thank the reviewer for pointing out this discussion, which gives some additional theoretical insights and points out better direction to be investigated within the CTDG domain. We added this analysis to discuss the experimental results (cf. the first paragraph of Section 4.2).
>
> For JODIE, when subject to a new event, the model only updates the nodes that are involved while the other nodes only benefit from the temporal encoding. Therefore, when the events are stochastic, the temporal encoder fails to capture the right next event. This effect mainly can be seen in Table 1 (as we agree with the reviewer), where JODIE clearly underperforms the other benchmarks, since the dataset requires both some long-range dependencies and also doesn’t have any temporal consistency. In Table 3, the community-based topology results in all the benchmarks (TGN, TGAT and JODIE) to fail. Meanwhile, we can see that while JODIE out-performs a bit TGAT, this can be simply explained by the construction choices of the SBM dataset (Table 4 - Appendix G), where we consider the existence of a probability that controls the occurrence of the events. We estimate that this probability gives a certain temporal occurrence that is partly well tracked by JODIE and therefore can still give some acceptable results (in comparison to TGAT). We added this discussion in the experimental part (cf. changes to the second and last paragraph in Section 4.2).

---

> > ### Comment · Reviewer_8yma · 2025-01-13
> >
> > Dear authors,
> >
> > thanks for incorporating my suggestions so far. Figure 3 indeed does look better. I'm also satisfied with the changes in Sections 2.2 and 4.2. I think there remain some unclear points, i.e., an analysis of how over-smoothing and IF interact, but I think they can be solved by future works, as enough material is presented in the current manuscript.
> >
> > As long as the table and callouts (from rebuttal part 2/3 of the authors) are added in the final version, the authors resolved my concerns, and I thank them for the interesting discussion.

---

> > > ### Author Response · Authors · 2025-01-13
> > > **Thank you!**
> > >
> > > Dear Reviewer,
> > >
> > > Thank you for your thoughtful feedback and for acknowledging the improvements in Figure 3 and Sections 2.2 and 4.2. We appreciate your suggestion regarding the interaction between over-smoothing and IF and agree that it presents an interesting direction for future research.
> > >
> > > We will ensure the table and callouts from rebuttal part 2/3 are included in the final version, as suggested. Thank you for your valuable insights and engaging discussion throughout the review process.
> > >
> > > Best regards,
> > > Authors of TMLR submission 3845

---

### Author Response · Authors · 2025-02-25
**Paper Revision based on discussion with the reviewers**

We thank all the reviewers for their comments, questions and insights that helped enhance our manuscript. Based on the different suggestions provided by the reviewers, we have updated the manuscript and we have uploaded a new version. Specifically, we have:

- Expanded the discussion on the theoretical assumptions we make in our analysis (Appendix A). We additionally included some elements of clarity in the different proofs (Appendix B and C).
- Provided the code to reproduce the results in the supplementary materials, specially for the synthetic data we created to showcase the effect of SBM-type graphs. The code is included in the supplementary materials, and will be publicly available upon publication.
- Increased the clarity on the proposed Information-Flow framework and provided a visual summary in Appendix F (based on discussion with "Reviewer 8yma").
- Provided additional results on "TGBL-Coin" dataset (Table 7 - Appendix G), and consequently covering multiple types of graphs.

We hope these modifications address the different comments and we are happy to engage in further discussion to improve our manuscript if needed.

---

### Decision · Action_Editor_8kCz · 2025-03-04

**Recommendation:** Accept as is

**Comment:**

The paper introduces a novel theoretical framework for analyzing continuous-time dynamic graphs (CTDGs) and provides empirical validation on diverse datasets. The claims are well-supported by evidence, and the work is relevant to TMLR’s audience. The paper offers valuable insights into graph representation learning in dynamic settings and highlights the potential of self-supervised learning approaches. Overall, the work is sound and makes a meaningful contribution to the field.

**Audience:**

The paper is relevant to TMLR’s audience. It addresses a significant gap in dynamic graph representation learning and provides a roadmap for future research, making it of interest to both theoretical and applied researchers.

**Claims And Evidence:**

The claims are supported by clear evidence. The paper introduces a theoretical framework for analyzing continuous-time dynamic graphs (CTDGs) and validates it empirically on diverse datasets. The results demonstrate the strengths and limitations of different methods, supporting the claims.